# Bridging the gap between the evolutionary dynamics and the molecular mechanisms of meiosis: A model based exploration of the *PRDM9* intra-genomic Red Queen

**Alice Genestier**, **Laurent Duret**, **Nicolas Lartillot** *

Universite Claude Bernard Lyon 1, LBBE, UMR 5558, CNRS, VAS, Villeurbanne, France

* nicolas.lartillot@univ-lyon1.fr

**Data Availability Statement:** The model, and the codes to generate the figures can be found at https://github.com/alicegenestier/Red_Queen_

## Abstract

Molecular dissection of meiotic recombination in mammals, combined with population-genetic and comparative studies, have revealed a complex evolutionary dynamic characterized by short-lived recombination hotspots. Hotspots are chromosome positions containing DNA sequences where the protein PRDM9 can bind and cause crossing-over. To explain these fast evolutionary dynamic, a so-called intra-genomic Red Queen model has been proposed, based on the interplay between two antagonistic forces: biased gene conversion, mediated by double-strand breaks, resulting in hotspot extinction (the hotspot conversion paradox), followed by positive selection favoring mutant *PRDM9* alleles recognizing new sequence motifs. Although this model predicts many empirical observations, the exact causes of the positive selection acting on new *PRDM9* alleles is still not well understood. In this direction, experiment on mouse hybrids have suggested that, in addition to targeting double strand breaks, *PRDM9* has another role during meiosis. Specifically, PRDM9 symmetric binding (simultaneous binding at the same site on both homologues) would facilitate homology search and, as a result, the pairing of the homologues. Although discovered in hybrids, this second function of *PRDM9* could also be involved in the evolutionary dynamic observed within populations. To address this point, here, we present a theoretical model of the evolutionary dynamic of meiotic recombination integrating current knowledge about the molecular function of PRDM9. Our modeling work gives important insights into the selective forces driving the turnover of recombination hotspots. Specifically, the reduced symmetrical binding of PRDM9 caused by the loss of high affinity binding sites induces a net positive selection eliciting new *PRDM9* alleles recognizing new targets. The model also offers new insights about the influence of the gene dosage of PRDM9, which can paradoxically result in negative selection on new *PRDM9* alleles entering the population, driving their eviction and thus reducing standing variation at this locus.

PRDM9_Panmictic.git or on Zenodo with all the datasets. doi:10.5281/zenodo.10840159.

**Funding:** This work was funded by the Agence Nationale de la Recherche, Grant ANR-19-CE12-0019 / HotRec. AG has a PhD scholarship contributed by the French Ministry of Research and Higher Education. The funders had no role in study design, data collection and analysis, decision to publish, or preparation of the manuscript.

**Competing interests:** The authors have declared that no competing interests exist.

## Author summary

Meiosis is an important step in the eukaryotic life cycle, leading to the formation of gametes and implementing genetic mixing by recombination of paternal and maternal genomes. A key step of meiosis is the pairing of homologous chromosomes, which is required in order to distribute them evenly into the gametes. Chromosome pairing will also determine the exact position at which paternal and maternal chromosomes will exchange material. Research on the molecular basis of meiosis has revealed the role of a key gene, *PRDM9*. The protein encoded by *PRDM9* binds to specific DNA sequences, by which it determines the location of recombination points. Symmetric binding of the protein (at the same position on the homologous chromosomes) also facilitates chromosome pairing. This molecular mechanism, however, has paradoxical consequences, among which the local destruction of the DNA sequences recognized by PRDM9, leading to their rapid loss at the level of the population over a short evolutionary time. In order to better understand why recombination is maintained over time despite this process, we have developed a simulation program implementing a model taking into account these molecular mechanisms. Our model makes realistic predictions about recombination evolution and confirms the important role played by *PRDM9* during meiosis.

## Introduction

In eukaryotes, meiosis is a fundamental step in the reproduction process, allowing the formation of haploid cells from a diploid cell. This process requires the success of a key step, namely, the pairing of homologous chromosomes. Correct pairing is essential for proper segregation of chromosomes into daughter cells. In addition, it allows for the formation of cross-overs, thus implementing recombination and generating new combinations of alleles. Finally, meiosis and recombination are at the heart of the questions of hybrid sterility and speciation [1–3]. However, the evolutionary dynamics of meiotic recombination still remains poorly understood. A correct understanding of this dynamics requires explicit description of the population genetics processes, on one side, and the molecular mechanisms of meiosis, on the other side. In the present work, we present an attempt in this direction, using theoretical and simulation models.

In mammals and many other eukaryotes, recombination points are not uniformly distributed along the chromosomes [4]. Instead, crossovers frequently occur at the same positions in independent meioses, into regions of the genome called recombination hotspots, where the frequency of crossing-over occurrence is 10 to 100 times higher than in the rest of the genome [5, 6]. Recombination hotspots are typically 1 to 2 kb long, and they are often located outside of genes. More than 30,000 hotspots were found in humans [4, 7], and around 40,000 in mice [8]. These hotspots are characterized in humans by the presence of a sequence motif (13-mer) determining up to 40% of crossing-overs [9]. This motif has helped to identify the *PRDM9* gene as the gene responsible for hotspot location, [9, 10]. *PRDM9* encodes a DNA-binding protein, and the hotspots therefore correspond to strong binding sites of this protein.

Once bound to its target site, the PRDM9 protein trimethylates surrounding histones (H3K4me3 and H3K36me3 [11, 12]), inducing the recruitment of the double strand break (DSB) machinery near the target site [13] (for a review see [14]). DSB induction, followed by DNA resection, produces a single-stranded end that searches for its homologous sequence on the other chromosome and is then repaired, using the sequence at the same locus on the homologous chromosome as the template. This repair leads to the conversion of the sequence

located near the break (often overlapping the site targeted by PRDM9 [15]). Some of these gene conversion events lead to the formation of crossing-overs (CO), while others are repaired without exchange of flanking regions (non crossing-overs, NCO). In some cases, repair is done not with the homologue but with the sister chromatid [15].

Crucially, when allelic variation exists at a given target motif that modulates the binding affinity for PRDM9, DSBs will form more frequently on the allele for which PRDM9 has the highest affinity (the 'hot' allele). Given that DSBs are repaired by using the intact homologue as a template, this process leads to the preferential replacement of 'hot' alleles by alleles for which PRDM9 has a lower affinity. This mechanism of biased gene conversion favors the transmission of mutations that inactivate hotspots, leading to an increase in the frequency of these mutant inactive versions of binding sites, as well as their fixation in the population. This self-destruction phenomenon, commonly called the "hotspot conversion paradox" [16], leads to the progressive inactivation, hereafter called *erosion*, of PRDM9 binding sites at the genome scale, over short evolutionary times (in the order of 10,000 to 100,000 generations [17]), which therefore raises the question of the maintenance of recombination in the long term.

As a solution to the paradox, a model has been proposed [18], which works as follows: the disappearance of recombination hotspots compromises the proper functioning of meiosis and greatly reduces the fertility of individuals. In this context, new *PRDM9* alleles, recognizing new target sites already present by chance in the genome and thus restoring recombination, would be positively selected. They would eventually replace the alleles currently segregating in the population. By analogy with the so-called Red Queen dynamics [19] typically displayed by prey-predator or host-pathogen systems, this model has been called the intragenomic Red Queen model of recombination hotspot evolution [18]. It predicts a rapid evolutionary dynamic of recombination landscapes, as well as a strong positive selection on the DNA binding domain of *PRDM9*.

Several empirical observations support the Red Queen model. First, fine scale recombination landscapes differ between closely related species, like between humans and chimpanzees [10, 20], or between modern humans and Denisovan [17], suggesting a fast turnover of recombination hotspots. Second, the DNA binding domain of PRDM9 is a zinc finger domain, consisting of an array of 7 to 10 zinc fingers. This domain is encoded by a minisatellite, which mutates rapidly [21] by a combination of point mutations and unequal crossing over between the sequence repeats. This allows for the rapid accumulation of new combinations of zinc fingers, and thus of new alleles recognizing different hotspots, providing the necessary mutational input for the Red Queen to run. In part because of this high mutation rate, *PRDM9* is typically characterized by a high genetic diversity in natural populations [22–26]. Finally, non-synonymous substitutions in the zinc finger domain of *PRDM9* are more frequent than synonymous substitutions, suggesting the presence of a strong positive selection acting on the DNA binding domain [3, 24].

Studies combining computer simulations and theoretical analyses have also given many arguments in favor of the Red Queen model [18], [27]. However, current theoretical models remain silent on two essential points. First, what exact mechanism explains the positive selection acting on *PRDM9* to restore recombination? Current models invoke a decline in fertility caused by the erosion of recombination, but without providing a precise explanation of this point. Second, these models do not provide any explanation, at this stage, of the link between the intra-genomic Red Queen and the hybrid sterility seen when mice are heterozygous for certain *PRDM9* alleles. [1, 28].

In this respect, a series of molecular and comparative analyses conducted on the mouse provide some clues. First, the differential erosion of PRDM9 binding sites between mouse subspecies (due to different *PRDM9* alleles eroding their respective target sites within each

subspecies) leads to asymmetrical binding configurations in hybrid individuals [29]. Specifically, in a F1 hybrid, each of the two *PRDM9* alleles has eroded its targets in the genome of its parental strain, but not in the other strain's genome. Each *PRDM9* allele will therefore tend to bind preferentially to the still active target sites present on the chromosomes inherited from the other parent, but not to the homologous but eroded sites on the chromosome from its own parent. These asymmetrical binding patterns of PRDM9 across the whole genome are suspected to be involved in the sterility phenotype. Indeed, Chip-Seq experiments have uncovered a correlation between PRDM9 binding asymmetry rates in a variety of mouse hybrids for different pairs of *PRDM9* alleles and hybrid sterility [30, 31]. Cytogenetic observations have also shown that chromosomes that are more asymmetrically bound are less often correctly paired during metaphase I [31]. Finally, the introduction by transgenesis in mice of a *PRDM9* allele possessing the human zinc finger domain, which has never been present in the mouse species complex and has then not eroded its targets asymmetrically in either of the two populations, restores both a good binding symmetry and high levels of fertility [1].

In the light of these empirical results, a model has been proposed [1], according to which *PRDM9* would in fact have a dual role during meiosis: in addition to being responsible for the recruitment of the DSB machinery, it would also facilitate the pairing between the homologous chromosomes, thanks to its symmetrical binding, that is, its simultaneous binding to the same target site on both homologues. More precisely, PRDM9 is thought to act to bring the sites on which it is bound to the chromosomal axis. The mechanisms involved in this step are not very clear, although it would seem that H3K4me3 markers [32] and the SSXRD and KRAB domains of PRDM9 are implicated [33]. This is where symmetry would play a key role (Fig 1): when PRDM9 is bound symmetrically, the two homologous loci are each brought closer to the chromosomal axis, which would, by some yet unknown mechanism, make the complementary sequence of the homologue more directly accessible to the single-stranded end produced by resection of the DSB [34, 35]. In contrast, in the case where PRDM9 binds only on one of the two homologous loci, a possible DSB would be repaired only later on, either by the homologue as a non CO event or by the sister chromatid [15].

This mechanistic model based on the symmetrical binding of PRDM9 provides a globally coherent explanation of the hybrid sterility phenotype. A last question remains open however: could this dual role of *PRDM9* also be implicated in the Red Queen evolutionary dynamic of recombination observed within a single population? Of note, in the hybrid, failure of meiosis is a consequence of macroscopic asymmetric sequence patterns due to differential erosion in the two parental lineages. Such macroscopic asymmetric sequence patterns are unlikely within a single population. On the other hand, statistical binding asymmetries might nevertheless occur and play a role.

To investigate this question, in this paper, we revisit the theoretical modeling of the intra-genomic Red Queen of *PRDM9*-dependent recombination. In contrast to previous work [18, 27] (but see the recent work of Baker *et al.* [36]), we explicitly model the molecular mechanism of meiosis and, more specifically, the function currently hypothesized for *PRDM9*. Our specific aim was to investigate whether the combined effects of biased gene conversion and symmetry provide sufficient ingredients for running an intra-genomic Red Queen able to explain current empirical observations.

## Results

To investigate the evolutionary dynamic of *PRDM9*-dependent recombination, we developed a simulation program modeling the evolution of a randomly mating population of diploid individuals and accounting for the key molecular processes involved in meiotic recombination

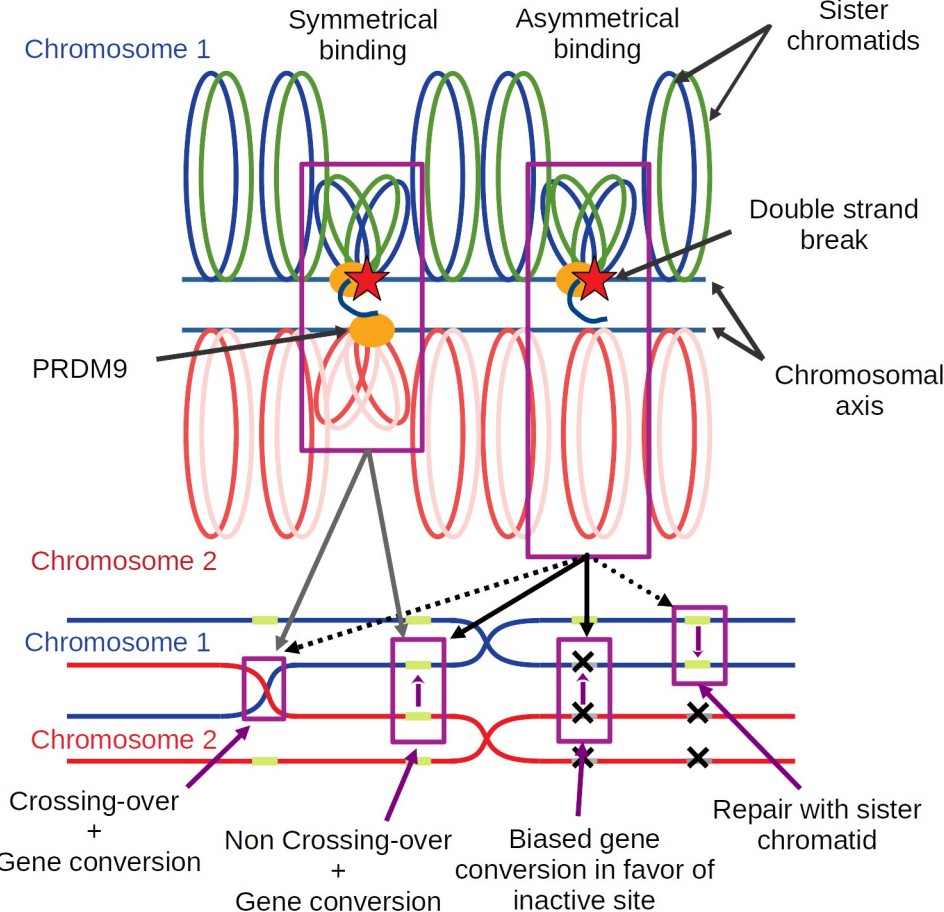

**Fig 1. A model of how symmetrical PRDM9 binding facilitates chromosome pairing.** Upon binding DNA at a specific target motif, PRDM9 (orange oval) brings the DNA segment close to the chromosomal axis. Some of the sites bound by PRDM9 may then undergo a DSB (red star). Resection of the DSB generates a single-stranded end, which will search for a complementary sequence to use as a template for repair. In the case where PRDM9 is symmetrically bound (i.e. on both homologues, case on the left-hand side), and assuming that homology search is restricted to the axis region, the templates provided by the two sister chromatids of the homologue are more directly accessible, thus facilitating homology search and pairing with the homologue. The break can then be repaired either as a CO or a NCO event, in both cases, implementing gene conversion at the broken site. In the case of asymmetrical PRDM9 binding (case shown on the right-hand side), the homologue is less directly accessible, preventing efficient homologue engagement. The broken site is assumed to be repaired later on, once the homologues have synapsed (and this, thanks to other DSBs occurring at symmetrically bound sites somewhere else on the same pair of chromosomes), as NCO events. In the case where the homologue bears an inactive binding site at the position corresponding to the DSB, the NCO will effectively implement biased gene conversion in favor of the inactive version.

(Fig 2). The genome consists of a single chromosome, bearing a locus encoding the *PRDM9* gene (Fig 2A). The locus mutates at a rate *u*, producing new alleles recognizing different sequence motifs (Fig 2B). Each allele recognizes a set of binding sites randomly scattered across the genome, of varying binding affinity. Binding sites undergo inactivating mutations at a rate *v* (Fig 2B). At each generation, sexual reproduction entails the production of gametes that are themselves obtained by explicitly implementing the process of meiosis on randomly chosen individuals (Fig 2C–2F).

An individual is randomly chosen to attempt meiosis and reproduce. Meiosis is modeled step by step. First, the PRDM9 proteins bind to their target sites, according to a simple

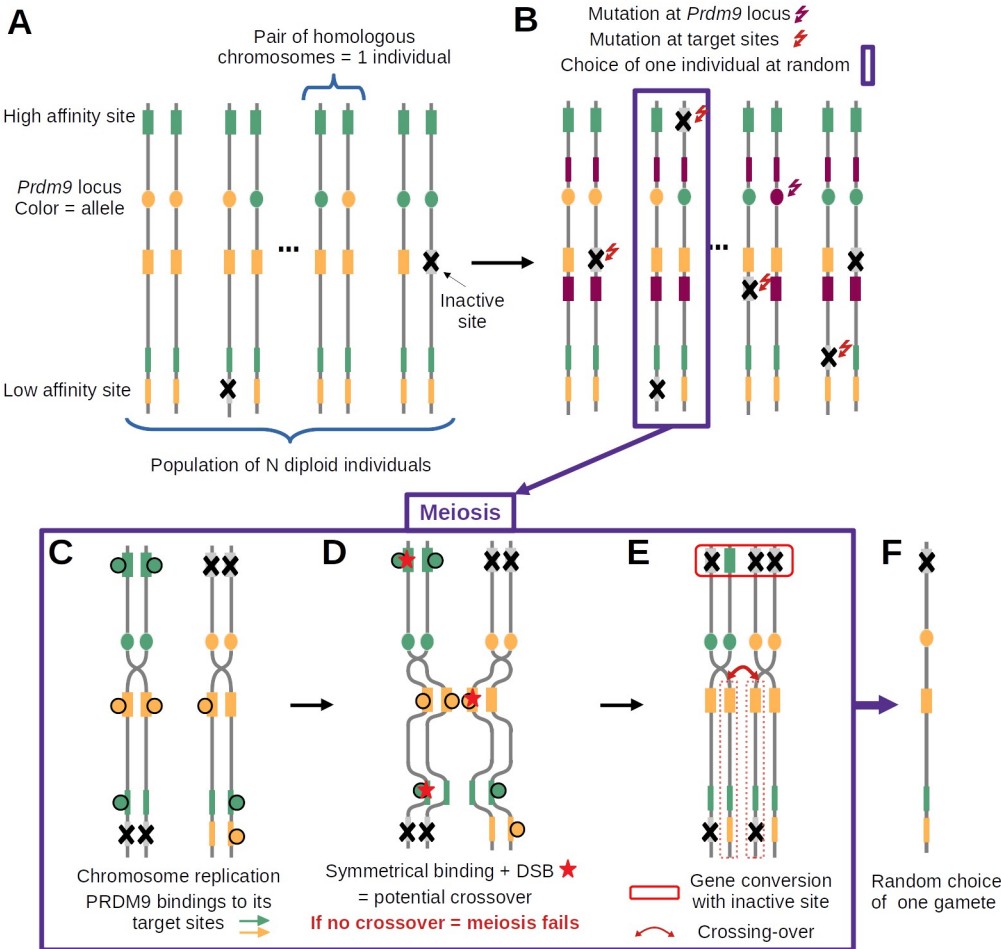

**Fig 2. Diagram summarizing the main features of the model and the successive steps of the simulation cycle.** (A) The model assumes a population of *N* diploid individuals (2*N* chromosomes, vertical lines). Each chromosome has a *PRDM9* locus (filled oval, with a different color for each allele) and, for each *PRDM9* allele, a set of target sites (filled rectangles, with color matching their cognate *PRDM9* allele) of variable binding affinity (variable width of the filled rectangles). (B) Mutations at the *PRDM9* locus create new alleles (here, purple allele), while mutations at the target sites inactivate PRDM9 binding (grey sites with a cross). (C-E) meiosis (here in a heterozygous individual); (C) Chromosome replication and binding of PRDM9 to its target sites (filled circles). (D) Induction of DSBs at a small number of randomly chosen sites bound by PRDM9 (red stars) and search for DSBs at symmetrically bound sites (symmetrical DSBs); if no symmetrical DSB is found, meiosis fails; otherwise, one symmetrical DSB is chosen uniformly at random (here, on the yellow binding site). (E) Completion of the crossing-over (red curved arrow) and repair of all DSBs using the homologous chromosome (red box on top); (F) random choice of one of the 4 gametes of the tetrad, which will fuse with another gamete and become part of the next generation.

chemical equilibrium (Fig 2C). The probability of occupation of a site will thus depend on the binding affinity of this hotspot site for PRDM9. It may also depend on PRDM9 concentration, which can itself depend on the genotype of the individual (either homozygous for single *PRDM9* allele, or heterozygous for two distinct *PRDM9* alleles), through gene dosage effects. Then, a small number of double strand breaks are induced at some of the sites bound by PRDM9 (Fig 2D). At that step, a key assumption of the model is that meiosis will succeed only if at least one of those DSBs occurs at a site that is also bound by PRDM9 on at least one of the two chromatids of the homologous chromosome, in which case a single cross-over is performed at the site of one of the DSBs fulfilling this requirement. Finally, all DSBs are repaired

using the homologue (Fig 2E), upon which meiosis is assumed to proceed successfully, producing four gametes, one of which is chosen at random for reproduction (Fig 2F).

Of note, because of the symmetry requirement, the probability of success of meiosis will depend on the genotype of the individuals on which it is attempted, which will thus induce differences in fertility between individuals. In addition, explicit repair of the DSBs by the homologue effectively implements the process of gene conversion at the binding sites, causing hotspot erosion in the population. The main question is then to what extent these two aspects of the molecular mechanism will influence the evolutionary dynamic.

The overall procedure is run for several tens of thousands of generations, during which several summary statistics are monitored: the frequency of each *PRDM9* allele and the resulting genetic diversity at the *PRDM9* locus, the mean proportion of binding sites of *PRDM9* alleles that are currently active, their mean affinity across the genome, the mean probability of symmetrical binding and the mean probability of success of meiosis. The model parameters and the monitored statistics are summarized in Table 1

The results section is divided in three main parts. First, a simple version of the model is presented, in which PRDM9 gene dosage is ignored (that is, assuming that the number of proteins produced by a given *PRDM9* allele is the same in a homozygote and in a heterozygote for this allele). Of note, this model is not empirically plausible. It implies that the total concentration

**Table 1. Description of input parameters and output variables.**

| Parameters | Description | Value |
|---|---|---|
| $u$ | Mutation rate at the *PRDM9* locus | $2 \times 10^{-6}$ to $5 \times 10^{-3}$ |
| $v$ | Mutation rate at the targets | $2 \times 10^{-6}$ to $5 \times 10^{-3}$ |
| $N$ | Population size | 5, 000 |
| $h$ | Number of targets recognised by a new allele | 400 |
| $d$ | Mean number of DSB per chromosome pair per meiocyte | 6 |
| $y_i$ | Affinity of target $i$ | variable |
| $g$ | Gene conversion rate | variable |
| $n_{mei}$ | Number of meioses allowed per individual before reproduction failure | 1 to 5 |

| Variables | Description | |
|---|---|---|
| $f_i$ | Frequency of allele $i$ [a] | |
| $\theta_i$ | Proportion of target sites still active for allele $i$; erosion level is then defined as $1 - \theta_i$ [a] | |
| $q_i$ | Mean probability of symmetrical binding of allele $i$ [a,b] | |
| $w_i$ | Mean fertility of allele $i$ ($\bar{w}$ is the mean over all of the alleles) [a,b] | |
| $\rho$ | Net rate of erosion per generation | |
| $x_j$ | Occupancy probability at site $j$ at equilibrium | |
| $z$ | Intrinsic age of a generic allele (Eqs 2 and 3)[a] | |
| $\bar{z}$ | Mean intrinsic age—also an analytical estimate of mean erosion level (Eqs 3 and 4)[a] | |
| $\alpha$ | Linear response of the log-fitness as a function of erosion | |
| $\tau$ | Mean time between successive invasions of the population by new *PRDM9* alleles | |
| $D$ | *PRDM9* diversity in the population at equilibrium | |
| $s_0$ | mean selection coefficient acting on new *PRDM9* alleles at equilibrium | |
| $\sigma_0$ | Relative difference in fertility between homo- and heterozygotes (or equivalently, between homo- and hemizygotes) for young alleles | |
| $\bar{\sigma}$ | Mean relative difference in fertility between homozygotes and hemizygotes in the population | |

[a] These variables also change over time

[b] the mean is over all individuals carrying this allele (with a weight of 1 for heterozygotes and a weight of 2 for homozygotes).

of PRDM9 protein in meiocytes is twice as high in heterozygotes compared to homozygotes, something that would be possible only if there was an allele-specific transcriptional feedback. The fast turnover of *PRDM9* alleles, each with its own binding specificity, does not seem to be compatible with such a mechanism. Nevertheless, in spite of its lack of empirical relevance, this model represents a useful stepping stone for understanding the consequences of erosion and of the requirement of symmetrical binding, before considering the additional complications brought about by gene dosage. Then, in a second part, we introduce PRDM9 gene dosage and work out its consequences, by systematically comparing the outcome of the simulations run with and without dosage, under otherwise identical settings. Finally, we attempt an empirical calibration of the full model (including dosage) based on current knowledge in the mouse, so as to test its predictions.

### Intragenomic Red Queen

We first ran the model without gene dosage, and with a low mutation rate at the *PRDM9* locus ($u = 5 \times 10^{-6}$). Note that the parameter values used here are not empirically relevant. Instead, the aim is to illustrate the different regimes produced by the model. An empirical calibration of the model will be presented below.

Assuming a low mutation rate for *PRDM9* results in few, rarely more than one, alleles segregating at any given time in the population (Fig 3A). We call this a *monomorphic* regime, after Latrille *et al.* [27]. The dynamic is as follows. First, an allele appears in the population and invades, until reaching a frequency close to 1. Meanwhile, the proportion $\theta$ of active binding sites for this allele decreases, due to inactivating mutations that are then driven to fixation by biased gene conversion Once the allele has eroded a fraction of around 20% of its sites, it is quickly replaced by a newly arisen *PRDM9* allele that recognizes a different hotspot sequence motif. This rapid replacement clearly suggests the presence of strong positive selection. The newly invading allele then erodes its target until being replaced by a new allele which in turn follows a similar trajectory, and so on.

A key aspect of the model is that it does not explicitly invoke a fitness function. Instead, the positive selection that seems to be operating on new alleles (Fig 3) is an emerging property of the mechanism of meiosis. This positive selection can be more precisely understood by examining how the probability of PRDM9 symmetrical binding (defined by Eqs (13) and (14), see Materials and methods section) and the mean fertility of a typical allele evolve over the lifespan of this allele (Fig 3D and 3E), and how this relates to the level of erosion (defined as the fraction of targets that have been inactivated), or equivalently the proportion of still-active targets (Fig 3B) and the mean affinity of the remaining non-eroded sites (Fig 3C).

First, we observe a clear correlation between allele frequency and proportion of active (non-eroded) target sites. When the frequency of an allele increases in the population, the proportion of still active sites for this allele decreases. This erosion (Fig 3B) seems to occur at a rate which is directly proportional to the frequency of the allele in the population (Fig 3A). This is expected: the more frequent an allele, the more often it will be implicated in a meiosis and the more opportunities its target sites will have to undergo gene conversion events.

Second, erosion results in a decrease of the mean affinity of the sites that are still active (Fig 3C). This reflects the fact that sites of high affinity are more often bound by PRDM9, and thus are more often converted by inactive mutant versions.

Third, sites of lower affinity are also less often symmetrically bound (i.e. bound simultaneously on both homologues). The key quantity that captures this effect is the conditional rate of symmetrical binding ($q$). Since DSBs are chosen uniformly at random among all bound sites by PRMD9, $q$ corresponds to the probability that a DSB will occur in a symmetrically

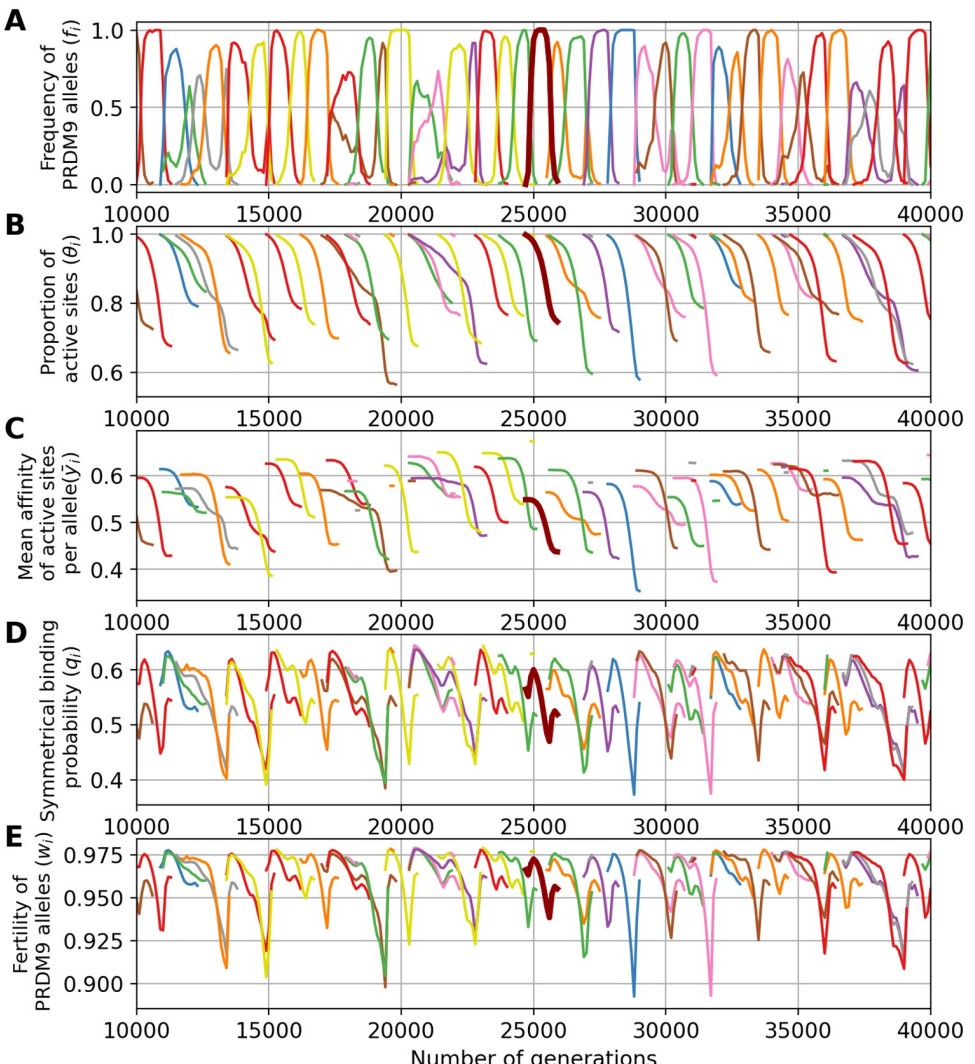

**Fig 3. A simulation trajectory showing a typical evolutionary dynamic, under a monomorphic regime (N = 5000, u = 5 × 10⁻⁶, v = 5 × 10⁻⁵).** In all panels, each color corresponds to a different allele. Note that a given color can be reassigned to a new allele later in the simulation. Each panel represents the variation through time of (A) the frequency of *PRDM9* alleles and their corresponding (B) proportion of active sites, (C) mean affinity, (D) probability of symmetrical binding and (E) fertility. The thick line singles out the trajectory of a typical allele.

bound site and will thus contribute to a successful pairing of the homologues. This probability is a monotonous function of the mean affinity of the binding sites for the PRDM9 protein (Fig 3D). The mean value of $q$ over all target sites of a given *PRDM9* allele is thus lower for older alleles (Fig 3D).

Finally, since, in our model, meiosis requires at least one DSB in a symmetrically bound site, the mean fertility of an older allele across all the meiosis it participates in is lower (Fig 3E). Hence, new alleles (young alleles) will be positively selected at the expense of old alleles and will ultimately replace them in the population.

To assess to what extent the requirement of symmetrical binding provides the selective force driving the evolutionary dynamics of *PRDM9*, we performed additional simulations with a simpler model, in which PRDM9 is required for the formation of DSBs, but symmetric

binding is not required for chromosome pairing. With this setting, the model still predicts a turnover of *PRDM9* alleles, but with high levels of erosion (36% of mean target inactivation under the control simulation, versus 12% under the model requiring symmetrical binding). Fundamentally, *PRDM9* alleles persist in the population until they have no more sites to bind (see S2 Fig), at which point they cannot anymore recruit DSBs and are thus selectively eliminated. Thus, a key result obtained here is that the requirement of symmetrical binding for chromosome pairing, combined with preferential erosion of high affinity sites, is sufficient for creating differences in fitness (fertility) between young alleles and older alleles having intermediate levels of erosion, ultimately providing the selective force behind the turnover of *PRDM9* alleles.

Of note some stochastic deviations from this typical life-cycle for a *PRDM9* allele are sometimes observed, such as an allele being first outcompeted by a subsequent allele but then showing a rebound in frequency when the competitor has itself eroded a large fraction of its target sites. Such deviations are relatively rare and do not seem to fundamentally change the overall regime. It should also be noted that we observe an increase in the rate of symmetrical binding and in the mean fertility at the beginning and at the end of the life of the alleles. The reason for this is that these two summary statistics are defined, for each allele, as a mean over all diploid genotypes segregating in the population that are carrying this allele. As a result, when old alleles have declined to a low frequency, they often find themselves in a heterozygous state with new alleles, which restores the rate of symmetrical binding and thus the fertility of the corresponding diploid individual. Likewise, when a new allele appears in the population, it is in a heterozygous state with an older allele, which gives a lower rate of symmetrical binding and fertility than being in the homozygous state.

The simulation shown above (Fig 3) was run under a low mutation rate, hence resulting in a *monomorphic* regime. Running the simulation under higher mutation rates for *PRDM9* ($u = 5 \times 10^{-4}$) results in a *polymorphic* regime, where many alleles segregate together at any given time. In this regime, of which a typical simulation is shown in Fig 4, the Red Queen process is also operating, except that many alleles are simultaneously segregating (Fig 4A), at a lower frequency. As in the previously shown monomorphic regime, each allele undergoes erosion (Fig 4B), primarily of its higher affinity sites (Fig 4C), again causing a decrease in symmetrical binding rate (Fig 4D) and fertility (Fig 4E). Owing to the high mutation rate, however there is a much lower erosion, leading to a lower decrease in fertility than in a monomorphic regime. Note that in Fig 4, the scale of the axes of the ordinates are not the same as in Fig 3 (see S1 Fig for a figure with same scale on the y-axis).

Control simulations (without the requirement of symmetry) run in the polymorphic case result in erosion levels of around 15% of the targets inactivated on average (see S2 Fig), thus not as high as in the control simulations in the monomorphic regime, although still much higher than in the simulations in which symmetrical binding is required (around 4%, Fig 4). Without the requirement of symmetry, and in a context where the number of sites bound by PRDM9 is in excess compared to DSBs (thus unlike the control shown above for a low mutation rate *u*), there is no selective difference between individuals. The overall dynamic under this control is therefore neutral, with a turnover caused by the mutational input of new alleles and loss of old alleles by drift. This is in contrast with the fundamentally selective regime observed under the requirement of symmetrical binding (with a difference in fertility between young and old alleles of the order of $10^{-3}$, Fig 4, bottom panel, resulting in a mean scaled selection coefficient in favor of new alleles of $4Ns_0 \simeq 34$). Thus, here also, as in the monomorphic regime considered above (Fig 3), the requirement of symmetrical binding, and more precisely the differential success of new and old alleles in fulfilling this requirement, is ultimately the mechanism mediating the selective force behind the turnover of *PRDM9* alleles.

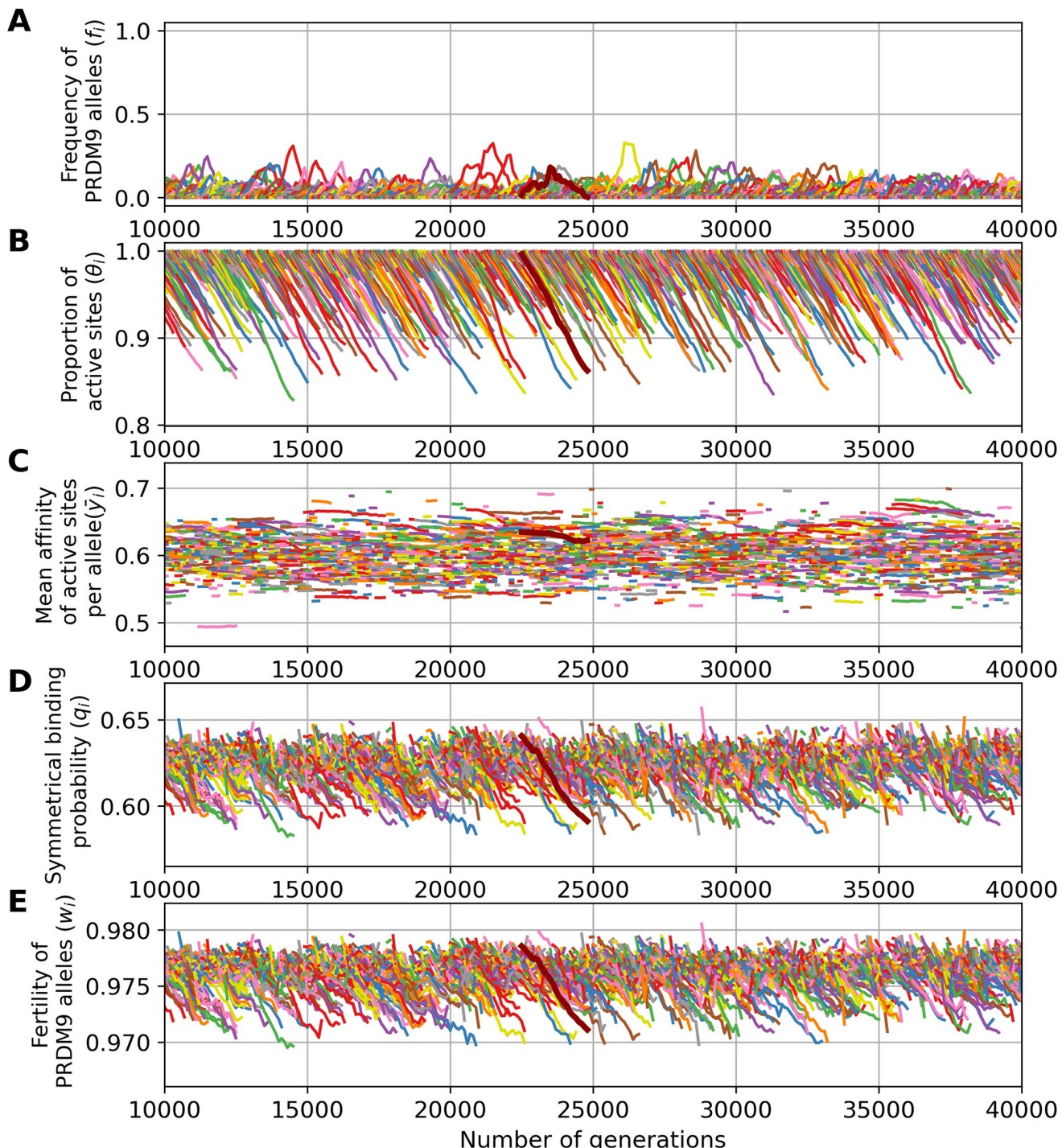

**Fig 4. A simulation trajectory showing a typical evolutionary dynamics, under a polymorphic regime ($N = 5000$, $u = 5 \times 10^{-4}$, $v = 5 \times 10^{-5}$).** In all panels, each color corresponds to a different allele. Note that a given color can be reassigned to a new allele later in the simulation. Successive panels represent the variation through time of (A) the frequency of *PRDM9* alleles and their corresponding (B) proportion of active sites, (C) mean affinity, (D) probability of symmetrical binding and (E) mean fertility. The thick line singles out the trajectory of a typical allele. Note that the scale of the y-axis for all panels is not the same as in Fig 3.

## Taking into account *PRDM9* gene dosage

The previous results were obtained with a model assuming the same concentration of the PRDM9 protein product of a given allele in individuals that are either homozygous or heterozygous for this allele. Yet, as mentioned above, this model is not realistic. Instead, gene dosage is likely to be an important aspect of the regulation of *PRDM9* expression [37]. To account for this fact, gene dosage was introduced in the simulation model. This was done by assuming that the amount of protein produced by a given allele is proportional to its copy number (i.e. twice as high in a homozygote than in a heterozygote). The main consequence of introducing gene dosage is that, in homozygotes, the occupancy of a site of a given affinity is increased, compared to a heterozygote, leading to a higher probability of symmetrical binding and fertility. This turns out to have an important impact on the Red Queen dynamics.

To illustrate this point, Fig 5 shows the results obtained with gene dosage in an identical parameter configuration as in the simulation without gene dosage shown in Fig 4. We observe a drastic change of regime between the two settings (compare Figs 4 and 5). While the simulation without gene dosage gives a polymorphic regime with an average maximum frequency of 0.2, the simulation with gene dosage gives a more heterogeneous dynamic. Indeed, in this regime, some alleles can partially or even completely invade the population for certain period of time. During this period of domination by one allele, new alleles apparently cannot invade the population.

What happens here is that, due to gene dosage, homozygotes have a fitness advantage over heterozygotes. Since alleles at high frequency are more often present in a homozygous diploid

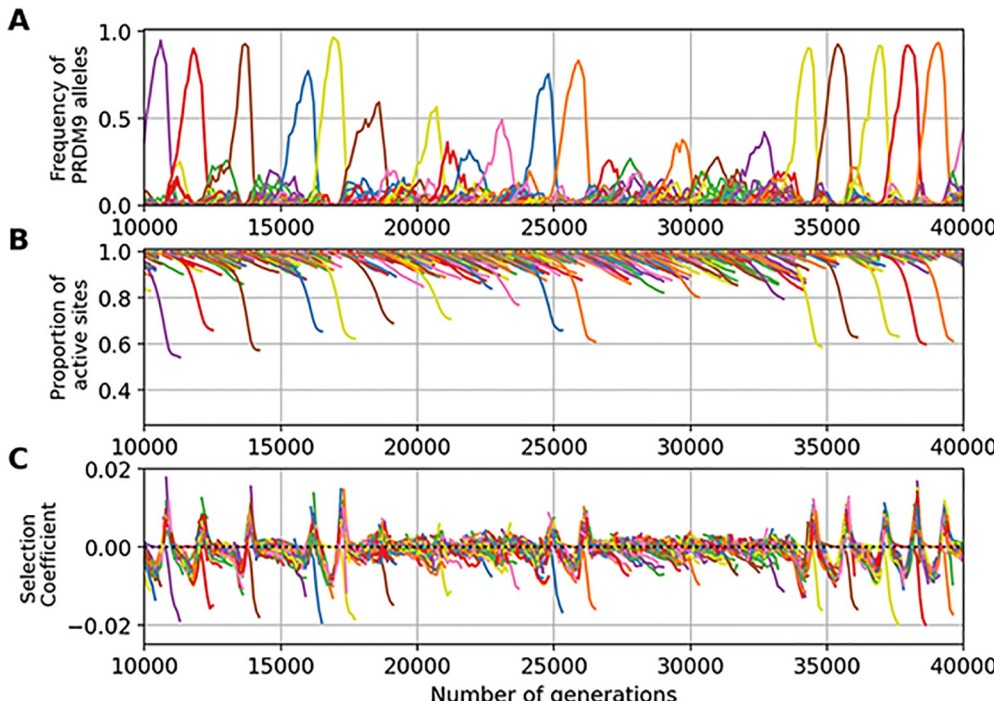

**Fig 5. Example of a *PRDM9* dynamic with gene dosage ($N = 5000$, mutation rates $u = 5 \times 10^{-4}$ at the *PRDM9* locus, $v = 5 \times 10^{-5}$ at the target sites).** In all panels, each color corresponds to a different allele. Note that a given color can be reassigned to a new allele later in the simulation. Successive panels represent the variation through time of (A) the frequency of *PRDM9* alleles and their corresponding (B) proportion of active sites and (C) selection coefficient acting on new alleles.

genotype, they have an advantage over low frequency alleles. And so, paradoxically, old (but not too old) alleles at high frequency can have a higher mean fitness than new alleles that just appeared in the population, and this, in spite of their higher levels of erosion. This last point is confirmed by measuring the selection coefficient associated with new alleles (Fig 5C): during a phase of domination of one allele, new alleles are strongly counter-selected. When the dominant allele becomes too old, its homozygote advantage is no longer strong enough to compensate for its erosion. At this point, new alleles become positively selected (e.g. at around 15, 000 generations on Fig 5C). The old allele then quickly disappears and all other alleles competes for invasion. This transient polymorphic regime is unstable however: as soon as one of the young alleles reaches a higher frequency than its competitors, it benefits from a homozogous advantage, ultimately eliminating all competitors from the population while still rare and thus becoming the new dominant allele. In the following, we call *eviction* this phenomenon fundamentally induced by gene dosage.

Running a control experiment without the requirement of symmetry (see S3 Fig) in the present case gives a regime characterized by higher levels of diversity ($D$ = 12, compared to $D$ = 7.5 when symmetrical binding is required), and an equilibrium erosion (15% of the targets inactivated on average) that is more pronounced than when symmetrical binding is required (8%). Again, as in the case without dosage, the regime is essentially characterized by a neutral turnover of *PRDM9* alleles. Thus, in the presence of dosage, the requirement of symmetrical binding to complete meiosis provides the selective force that drives both the selective replacement of old alleles by new ones (as in the case without dosage) and the eviction of all alleles except one (the specific contribution of dosage).

## Quantitative assessment of the equilibrium regime

The simulations shown in the previous section have helped to tease out the molecular details of the implication of *PRDM9* in the Red Queen dynamics of recombination. However, it remains to be understood how the equilibrium regime quantitatively depends on the parameters of the model. Another important question is to better understand the conditions under which gene dosage is expected to have an impact on the qualitative regime of the Red Queen, such as observed in the example shown above (Fig 5). To address these issues, scaling experiments were performed (see Methods and S4 Fig), systematically varying the parameters of the model so as to determine how the equilibrium changes accordingly, and systematically contrasting the models with and without dosage.

A first experiment was performed, varying the mutation rate $u$ at the *PRDM9* locus and the rate $v$ of target inactivation. The standing diversity at the *PRDM9* locus is strongly dependent on $u$, with higher diversity levels under higher values of $u$ (Fig 6A). This merely reflects the general fact that genetic diversity directly depends on the fresh mutational input at the locus. Diversity depends much more weakly on $v$, showing a variation of at most 20% over 4 orders of magnitude for $v$, except in a regime with otherwise intermediate values of $u$ ($4Nu$ = 10, 6C), where the equilibrium shows a sharp transition from a monomorphic ($D \simeq 1$) to a polymorphic ($D \simeq 10$) regime as $v$ is increased (Fig 6A).

Importantly, this sharp transition is critically dependent on the presence of gene dosage, being absent in the control conditions without gene dosage (Fig 6B). More generally, diversity tends to be lower, and the monomorphic regime more prevalent, in the presence of gene dosage (compare panels A and B). This confirms and generalizes the observations made above (Fig 5), while giving a better characterization of the conditions under which introducing gene dosage under otherwise constant parameter values can result in a transition from a

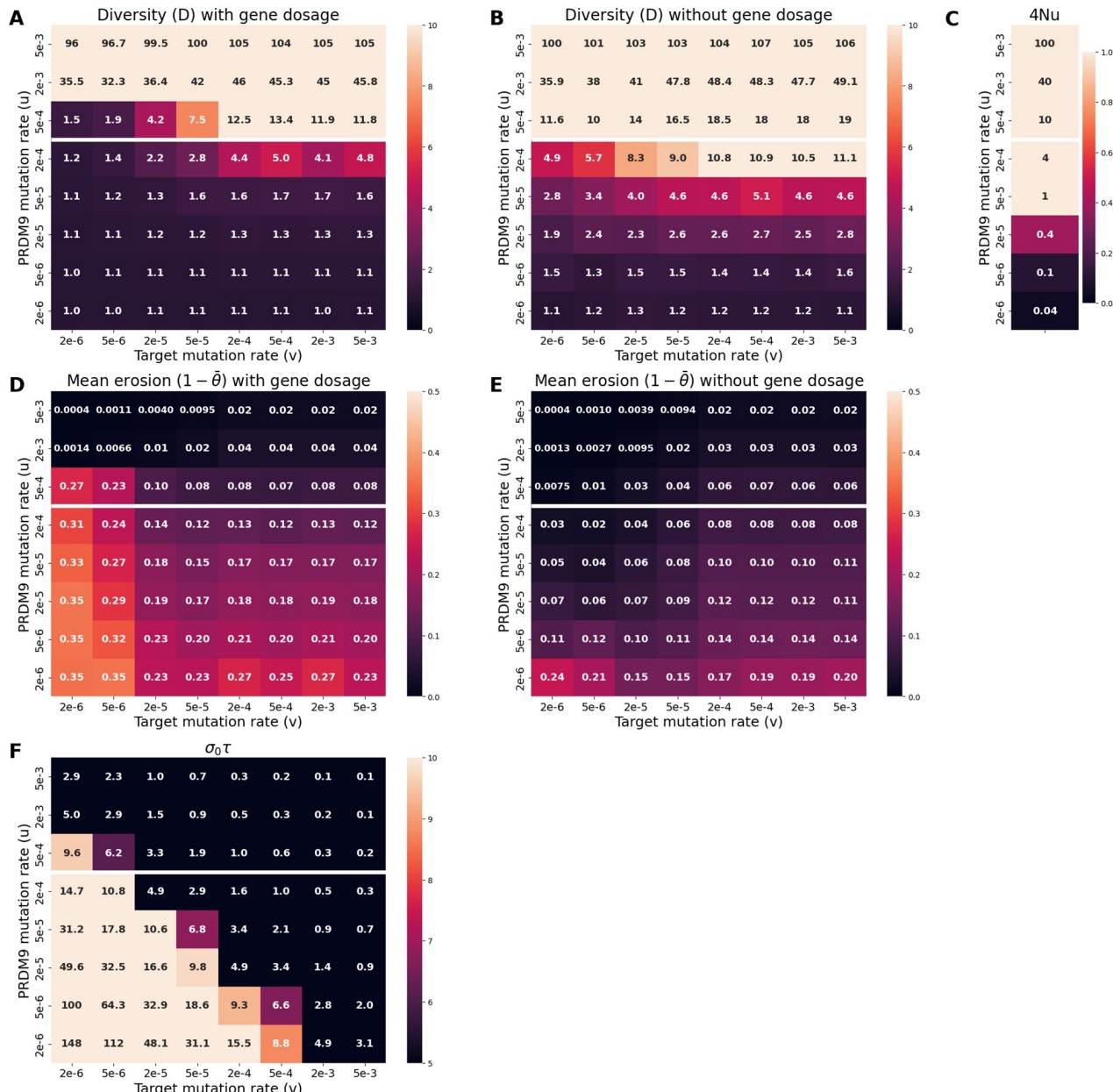

**Fig 6. Scaling experiment as a function of the mutation rates *u* and *v* (N = 5 × 10³).** A and B: equilibrium levels of *PRDM9* diversity with (A) and without (B) genetic dosage. D and E: equilibrium levels of *PRDM9* erosion level with (D) and without (E) dosage. C and F: summary statistics that are predictive of the type of regime observed in the presence of dosage, namely: (C) 4*Nu*, and (F) $\sigma_0\tau$ (selection coefficient associated to dosage multiplied by the time between successive invasions by *PRDM9* alleles). The white horizontal line separates the graph into two parts: 4*Nu* ≥ 10 above and 4*Nu* < 10 below. The regime is predicted to be polymorphic in the presence of dosage if 4*Nu* > 10 and $\sigma_0\tau$ < 3.

polymorphic to a monomorphic regime: essentially for values of 4*Nu* of the order of 10 and low values of v (Fig 6C).

When in the polymorphic regime (i.e. for high values of *u*, such that 4*Nu* ≥ 10, the mutation rates *u* and *v* have opposing effects on erosion levels (Fig 6D). This can be understood as follows. The equilibrium set point of the Red Queen is fundamentally the result of a balance between erosion and invasion. If *u* increases, this also increases the rate of replacement of old

alleles shifting the equilibrium towards weaker equilibrium erosion. Conversely, the rate of erosion is directly proportional to $v$, such that increasing $v$ will result in an increase of the rate of erosion at equilibrium and, correlatively, lower rates of symmetrical binding and fertility.

However, these opposing effects of $u$ and $v$ on equilibrium erosion levels are only valid for high values of $u$. For lower values of $u$ ($4Nu \leq 10$), on the other hand, as $v$ is increased, the equilibrium erosion level first shows a decrease and then stabilizes. This somewhat paradoxical response of erosion to changes in the rate of inactivation of the targets of PRDM9 is dependent on gene dosage: in the control experiment in which dosage is deactivated, the equilibrium erosion level always increases as a function of $v$, except for the lowest values of $u$ (Fig 6E).

## Characterization of the impact of gene dosage

The observations just presented show that gene dosage has a critical impact on the observed equilibrium. Specifically, for intermediate values of $u$ and low values of $v$, dosage tends to create a monomorphic regime characterized by high levels of erosion.

To better understand this point, an analytical approximation was derived, using the mean-field approach introduced in Latrille *et al* [27]. The complete analytical developments can be found in Sections A to C in S1 Appendix. Here, a simplified summary of the main result is given. This analytical derivation relies on several assumptions: a polymorphic regime, strong selection at the *PRDM9* locus, weak erosion, and a weak impact of gene dosage. It is thus not valid over the full range of parameter values explored here, in particular given the strong impact of dosage on the standing diversity of *PRDM9* observed above. However, it will give the conditions on the model parameters for which gene dosage is in fact not weak and is thus expected to have a strong impact on the observed dynamic.

First, to a good approximation, the rate of erosion of the targets of an allele depends on its frequency in the population:

$$\frac{d\theta}{dt} \approx -\rho f \theta, \tag{1}$$

where $\rho = \frac{Nvd}{2h} = 4Nvg$ and $g = \frac{d}{8h}$ is the net rate of erosion per generation. As a result, an analytical proxy for the cumulated erosion of an allele of age $t$ is given by:

$$z(t) = \rho \int_0^t f(t') \, dt', \tag{2}$$

where $f(t')$ is the frequency of the allele at time $t'$. The quantity $z$ can thus be seen as the *intrinsic age* of the allele (i.e. an allele ages more rapidly if more frequent in the population). Because of selection, the frequency of an allele changes as a function of its erosion, or intrinsic age $z$.

Assuming weak erosion ($z << 1$) allows one to linearize the differential equation describing the evolution of the frequency of an allele (see Section D in S1 Appendix), which gives:

$$\frac{d \ln f}{dt} \simeq -\frac{\alpha}{2}(z - \bar{z}) + \sigma_0(f - \bar{f}). \tag{3}$$

Here, $\bar{z}$ is the mean intrinsic age of all alleles currently segregating in the population, weighted by their frequency, and $\alpha$ is the slope of the log fitness as a function of erosion (how much fertility decreases as a function of the erosion of the targets of the allele). Thus, the first term of this equation captures the age effect: everything else being equal, older (and thus more eroded) alleles tend to bind less often symmetrically and therefore tend to have a lower fitness than younger allele. As a result, they will tend to undergo a decrease in their frequency in the population.

The second term of the equation is the specific contribution of gene dosage. In this second term, $f$ and $\bar{f}$ are the current frequency of the allele and the mean frequency of all other segregating in the population. As for $\sigma_0$, it is defined as the relative difference in fertility between homozygotes and heterozygotes for young alleles (with $z = 0$, see Eq 70 of Section D in S1 Appendix). It thus captures the homozygous advantage induced by gene dosage. The homozygous advantage depends on the age of the allele, however, under the weak erosion limit considered here, it can to good approximation be taken as constant and equal to that calculated for new alleles. Of note, under the assumptions of the model, $\sigma_0$ also measures the relative difference in fertility between a homozygote and a hemizygote for a new allele, that is, the haplo-insufficency of *PRDM9*.

This second term of the equation captures the frequency effect: everything being equal, more frequent alleles benefit from a homozygous advantage and will therefore tend to become even more frequent. Importantly, this term systematically acts against diversity. It captures the self-reinforcing nature of the homozygous advantage, and thus gives the key for explaining the origin of the eviction phenomenon observed in Fig 5.

Based on Eq 3, we can determine when dosage will play an important role. To address this point, we adopt a perturbative approach. Here, perturbative means that we first work out the equilibrium in the simpler situation where gene dosage is absent, thus ignoring the second term of Eq 3, and then characterize the conditions under which this term should in fact have an important impact.

In the absence of dosage, a self-consistent mean-field approximation for the mean equilibrium erosion can be obtained (see Sections B.4 and C.3 in S1 Appendix):

$$1 - \bar{\theta} = \bar{z} \approx \sqrt{\frac{vg}{u\alpha}}, \tag{4}$$

as well as the equilbirium diversity $D$ (see Section C.1 in S1 Appendix):

$$D \quad \approx \quad 24Nu. \tag{5}$$

Also of interest is the mean time $\tau$ between successive invasions by new *PRDM9* alleles (see Eq 21 in Section B.4 in S1 Appendix):

$$\tau = \frac{1}{2N} \frac{1}{\sqrt{u\alpha vg}}. \tag{6}$$

Note that these equations are the same as those presented in Latrille *et al.* [27]. However, in [27], $\alpha$ and $g$ were phenomenological parameters, set arbitrarily, while in our mechanistic model, they emerge directly from the molecular mechanisms of meiosis. As such, they can be expressed as functions of the model parameters ($d$, $h$, $N$ or $v$, see S1 Appendix Eq 59).

In Eqs 4 and 5, we recover a quantitative account of the main trends observed in the absence of dosage, such as explored above, namely, the opposing effects of $u$ (at the denominator) and $v$ (at the numerator) on the equilibrium erosion level (Eq 4), and the fact that the genetic diversity at the *PRDM9* locus mostly depends on $Nu$.

Based on these analytical results under the model without dosage, we can now determine when dosage will play an important role on standing diversity. First, the mutation rate should be high enough, so that diversity would be high ($D > 1$) if it were not for gene dosage, thus $Nu \gg 1$ (based on Eq 5). Second, the eviction effect (captured by the second term of Eq 3) should be sufficiently strong and have enough time to operate on the frequencies of the competing alleles before the replacement of those alleles by new mutant alleles. Quantitatively, eviction accumulates at a rate $\sigma_0$, over a typical time of $\tau$ (Eq 6), resulting in a cumulated effect

over the lifespan of an allele of the order of $\sigma_0\tau$. Thus, we predict a qualitative change induced by gene dosage on *PRDM9* standing diversity when $Nu$ is large and $\sigma_0\tau$ is also large. Conversely, we predict a polymorphic regime when $Nu$ is large and when the dosage effects are negligible, *i.e.* when $\sigma_0\tau$ is small. Finally, when $Nu$ is small, the regime is monomorphic irrespective of the strength of gene dosage.

These predictions were verified by systematically computing the mean value of $\sigma_0\tau$ across the scaling experiment on $u$ and $v$. In all cases where a pronounced difference is observed between 6A and 6B (essentially for large $u$ and small $v$), the criterion that $\sigma_0\tau$ should be large is met (*i.e.* is always greater than 3, Fig 6F). In the present case, $\sigma_0$ is not dependent on $u$ and $v$, only $\tau$ depends on these parameters. This explains why, for constant $u$, gene dosage has a stronger impact for low $v$: a low $v$ would imply a longer time between successive invasions (Eq 6), giving more time for the frequency effect (second term of Eq 3) to operate and thus pushing the Red Queen into the eviction regime.

In order to get additional insights on the impact of dosage, and more specifically on the role of $\sigma_0$, another bi-dimensional scaling experiment was performed, this time allowing for the variation of the parameters $d$, which is the mean number of DSBs per meiosis per chromosome pair, and $\bar{y}$, corresponding to the mean value of the binding affinity distribution of all the *PRDM9* target sites (Fig 7). Here, a change of regime is essentially found when these two parameters are low. It is also well predicted by the criterion that $\sigma_0\tau$ should be large (Fig 7C). Of note, both parameters, $d$ and $\bar{y}$, play on $\sigma_0$, rather than on $\tau$. Thus, if DSBs are limiting or sites tend to be of low affinity, a change in dosage can make a big difference in the chance of having at least one DSB in a symmetrically bound site.

These results show that gene dosage has important consequences on the Red Queen regime, by substantially restricting the conditions under which a high *PRDM9* diversity can be observed: not just a high mutation rate at this locus, but also a sufficiently short time between two invasions of alleles as well as a fairly weak heterozygous disadvantage (or, equivalently, a weak haplo-insufficiency) in young alleles.

## Empirical calibrations of the model

**Empirical input parameters.** Finally an empirical calibration of the model was attempted. The idea of this calibration is to try to match the parameters of the model to known empirical values, based on current knowledge in mammals and, more specifically, in the mouse, so as to

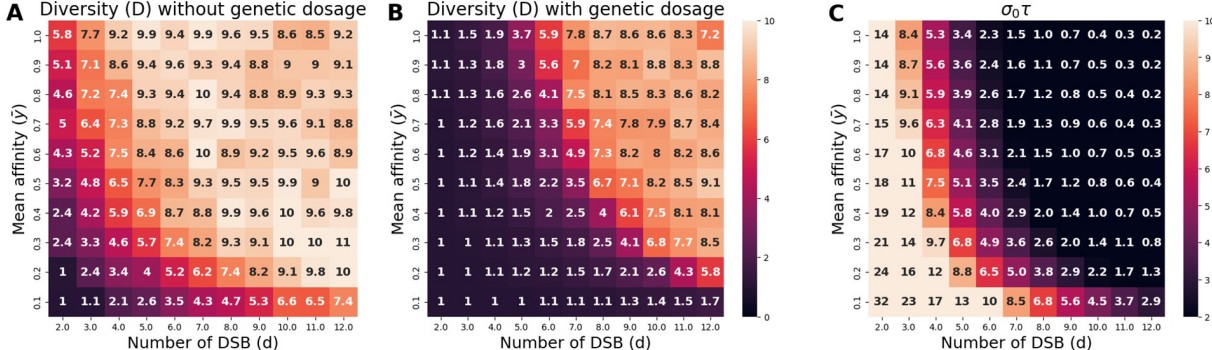

**Fig 7. Scaling experiment showing the change of regime (from polymorphic to monomorphic) caused by the introduction of genetic dosage (N = 5 × 10³).** A and B: equilibrium levels of *PRDM9* diversity without (A) and with (B) genetic dosage. C: summary statistic that is predictive of the type of regime observed in the presence of dosage, here $\sigma_0\tau$ (selection coefficient associated to dosage multiplied by the time between successive invasions by *PRDM9* alleles). The regime is predicted to be polymorphic in the presence of dosage if $\sigma_0\tau < 3$.

see whether the model is able to at least roughly reproduce key empirical observations, such as the typical erosion levels or genetic diversity at the *PRDM9* locus.

Focussing on the mouse, the effective population size ($N_e$) of different *Mus musculus* subspecies such as *Mus m. musculus* or *Mus m. domesticus* is around $10^5$ [38]. The number of hostspots recognized per *PRDM9* alleles across the genome ($h$) is estimated between 15, 000 and 20, 000 [39]. Knowing that the mouse possesses 20 pairs of chromosomes, this corresponds approximately to $h \approx 1000$ sites per chromosome. The mean number of DSBs ($d$) performed per individual is estimated between 200 and 250 per meiosis [40], which correspond to approximately 10 DSBs per chromosome pair per meiosis. An exponential distribution for the affinities has a good fit to observed distribution obtained from Chip-seq experiments [28] (see S5 Fig). The mean, however, is unknown but can be determined by assuming an average of 5, 000 targets bound per meiocyte [41] out of 20, 000 available targets. We obtain a mean of approximately 0.2.

Concerning the target mutation rate, considering that the mutation rate per nucleotide per generation across the mouse genome is $5.4.10^{-9}$ [42] and that the motifs characterising most of the hotspots in the mouse are several tens of nucleotides long [29], the inactivating mutation rate per generation ($v$) at the targets can be estimated at $10^{-7}$ [27]. The other mutation rate to determine is the one at the *PRDM9* locus ($u$). Owing to the mutational instability of the minisatellite encoding the zinc finger domain, this mutation rate is high and has been estimated at $10^{-5}$ in humans [21]. However, this is the raw mutation rate (including mutation that lead to either non-functional or functionally equivalent alleles). In contrast, the mutation rate $u$ of the model is the net functional mutation rate (probability of creating a new functional allele recognizing entirely different targets), and the net rate is likely to be smaller than the raw experimental estimate. Accordingly, and as in Latrille *et al.* [27], we used $u = 3.10^{-6}$.

Finally, since a population size of $N = 10^5$ is too large for running the simulator, a standard scaling argument was used, by setting $N = 5.10^3$ and multiplying $u$ and $v$ by a factor 20 (i.e. using between $u = 6.10^{-5}$ to $u = 6.10^{-4}$ and $v = 2.10^{-6}$). The other parameters are set for the smallest chromosome in the mouse, so with lower $h$ and $d$ than the mean in the entire genome ($h = 800$, $d = 8$, $\bar{y} = 0.2$). This rescaling leaves approximately invariant the following quantities: the PRDM9 diversity $D$, the erosion level $z$ or, equivalently, the proportion of active targets $\theta$, the homozygous advantage $\sigma$ and the selection coefficient experienced by new alleles $s_0$. The settings just presented represent our reference for empirical confrontation. Based on this reference, several variations of the model were also explored, which are described below.

**Model predictions.**   The simulator was calibrated based on these rough empirical estimates for its parameters, then run and monitored for several key summary statistics, specifically, the genetic diversity of *PRDM9* ($D$), the proportion of eroded sites ($1 - \bar{\theta}$), the mean haplo-insufficiency at the birth of the allele ($\sigma_0$), the mean haplo-insufficiency over alleles sampled at the equilibrium regime ($\bar{\sigma}$) and the selection coefficient experienced by new *PRDM9* alleles ($s_0$). The predictions are shown in Table 2 and examples of Red Queen dynamics are shown in Figs 8 and 9.

The first line of Table 2 reports the results obtained from simulations run under the parameter values corresponding to our reference for empirical comparison (see also Fig 8 for a typical simulation trajectory). The level of erosion ($1 - \bar{\theta}$) predicted by the model is consistent with what is known in the mouse (between 20% and 50% of erosion [1, 28, 29]). However, the predicted *PRDM9* diversity appears to be too low: these simulations result in a monomorphic regime (*i.e.* $D \sim 1$), whereas the *PRDM9* diversity observed in natural populations of mice is of the order of $D = 6$ to $D = 18$ [28] (see S1 Text). Of note, the diversity predicted by the model is the diversity of functionally different alleles (owing to the assumption made by our model of

**Table 2. Empirical calibration experiments.**

| $u$ [a] | $\bar{y}$ | $d$ | $c_{hom}$ | $n_{mei}$ | $D$ | $1-\bar{\theta}$ | $\sigma_0$ [b] | $\bar{\sigma}$ [b] | $s_0$ |
|---|---|---|---|---|---|---|---|---|---|
| $3\times10^{-6}$ | 0.2 | 8 | 2 | 1 | 1 | 0.25 | $2.9\times10^{-2}$ | $5.2\times10^{-2}$ | $-2.2\times10^{-2}$ |
| $3\times10^{-6}$ | 0.44 | 8 | 1 | 1 | 2.8 | 0.05 | $1.5\times10^{-5}$ | $-4.9\times10^{-7}$ | $5.5\times10^{-4}$ |
| $3\times10^{-6}$ | 0.3 | 8 | 1.5 | 1 | 1.1 | 0.22 | $1\times10^{-2}$ | $1.5\times10^{-2}$ | $-6\times10^{-3}$ |
| $3\times10^{-5}$ | 0.2 | 8 | 2 | 1 | 1.2 | 0.23 | $2.9\times10^{-2}$ | $4.6\times10^{-2}$ | $-2.2\times10^{-2}$ |
| $3\times10^{-6}$ | 2 | 8 | 2 | 1 | 2.1 | 0.23 | $5.1\times10^{-4}$ | $7.8\times10^{-4}$ | $-8.5\times10^{-5}$ |
| $3\times10^{-6}$ | 0.2 | 8 | 2 | 5 | 2.2 | 0.22 | $2.9\times10^{-2}$ | $6.9\times10^{-2}$ | $6.5\times10^{-6}$ |
| $3\times10^{-6}$ | 0.2 | 24 | 2 | 1 | 2.5 | 0.26 | $5.7\times10^{-5}$ | $3.3\times10^{-3}$ | $1.5\times10^{-4}$ |

Equilibrium values of output summary statistics at equilibrium (*PRDM9* diversity $D$, mean level of erosion $1-\bar{\theta}$, mean haplo-insufficiency at the birth of the allele ($\sigma_0$), mean haplo-insufficiency at the equilibrium ($\bar{\sigma}$) and the mean selection coefficient experienced by new *PRDM9* alleles ($s_0$) as a function of the input parameters (mutation rate at *PRDM9* locus ($u$), mean of the affinity distribution ($\bar{y}$), number of DSB par meiosis ($d$), dosage coefficient ($c_{hom}$) and maximum number of meiosis allowed for each individual), and with fixed values for the other parameters ($h = 800$ and $v = 10^{-7}$)

[a] The values reported are before the down-scaling of the model, for the original population size of $N = 10^5$ (thus $4Nu = 1.2$ when $u = 3\times10^{-6}$ and $4Nu = 12$ when $u = 3\times10^{-5}$)

[b] Haplo-insufficiency is here formally defined as the relative difference in success rate of meiosis between homozygotes and hemizygotes. This is equivalent to the relative difference in fertility, except in the case where $n_{mei} = 5$, where fitness and rate of successful meiosis are not proportional.

non-overlapping sets of targets for different alleles). In reality, closely related alleles can share a substantial fraction of their targets [28]. However, even accounting for this redundancy, the empirical functional diversity is still typically greater than 1, of the order of 2 to 6 in mouse subspecies [28] (see S1 Text). Thus, the monomorphic regime obtained here seems to be in contradiction with empirical observations.

This low diversity can be explained by the fact that we are in a range of parameters involving an eviction regime due to gene dosage effects (compare Figs 5A, 5B, 5C and 8A), such that one allele dominates in the population, while all the other alleles are counter-selected except during the short phases of allelic replacement (on average, the mean selection coefficient experienced by new alleles, $s_0$, is negative, Table 1).

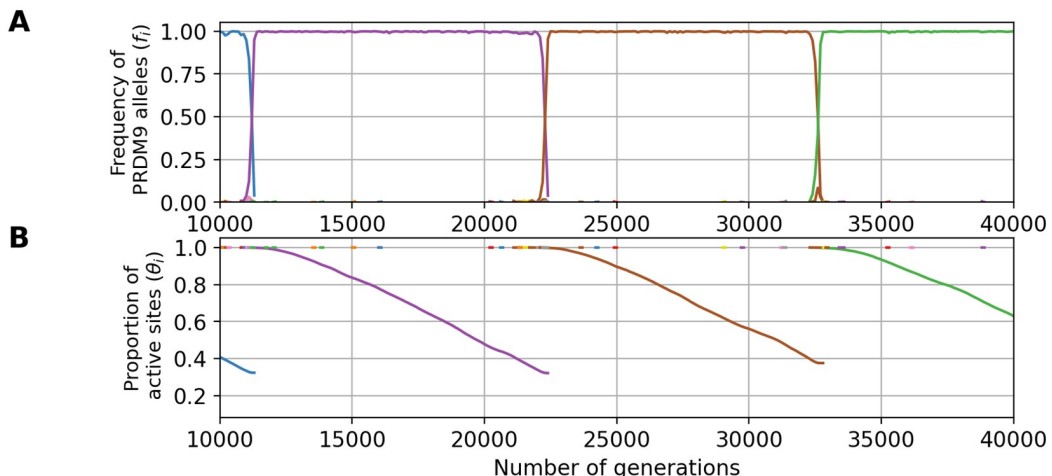

**Fig 8. An example evolutionary trajectory under the fitness scheme allowing for only one meiosis per individual ($n_{mei}$ = 1, and with parameters u = $3\times10^{-6}$, N = $10^5$ and $\bar{y}$ = 0.2).** Variation through time of the frequency of each *PRDM9* allele (A) and their proportion of active sites (B).

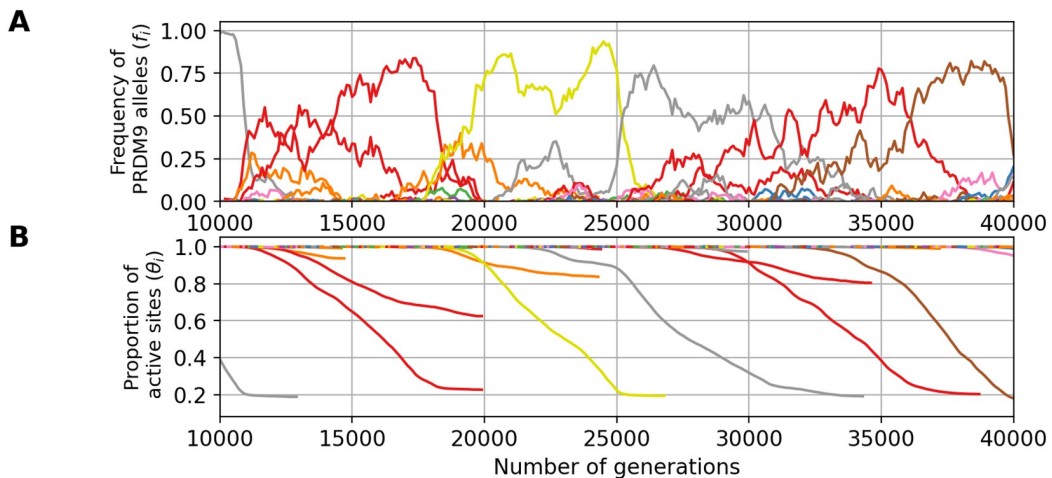

**Fig 9. An example evolutionary trajectory with 24 DSBs (d = 24, u = 3 × 10⁻⁶, N = 10⁵ and ȳ = 0.2).** Variation through time of the frequency of each *PRDM9* allele (A) and their proportion of active sites (B).

The fact that eviction is caused by gene dosage can be further verified by running the model without gene dosage (i.e. $c = 1$, Table 2, line 2), in which case higher levels of diversity are produced. Intermediate levels of gene dosage ($c = 1.5$, Table 2, line 3), on the other hand, give results that are essentially identical to those obtained with a dosage directly proportional to the number of gene copies ($c = 2$). Gene dosage also results in a non-negligible haplo-insufficiency, especially in old alleles (Table 2 column $\sigma\bar{z}$), with a reduction of a few percents in the success of meiosis in hemizygotes (or, equivalently, heterozygotes with two alleles of same age), compared to homozygotes. Interestingly, such levels of haplo-insufficiency are comparable to those observed in old alleles in the mouse (B6, C3H [37]). The predicted haplo-insufficiency of young alleles $\sigma_0$ is weaker (Table 2). Such an age-dependency for the impact of gene dosage was previously suggested [1]. Nevertheless, at least in its current form and under those parameter values, the model does not predict an empirically reasonable regime.

Increasing the *PRDM9* functional mutation rate ($u$) by as much as a factor 10 is not sufficient to get out of the eviction regime, resulting instead in a predicted *PRDM9* diversity still very close to one (fourth row of Table 2). Alternatively, increasing the mean affinity (or, equivalently, the concentration of PRDM9 in the cell) allows for higher levels of *PRDM9* equilibrium diversity while maintaining empirically reasonable levels of erosion (fifth row of Table 2). However, the mean number of sites bound by PRDM9 in a meiocyte would then predicted to be too large ($\sim 20,000$), compared to current empirical knowleldge ($\sim 5,000$ [41]). In addition, in this parameter regime, the model predicts very low levels of haplo-insufficiency ($\sigma_{\bar{z}} < 1 \times 10^{-3}$ for old alleles), substantially lower than empirically observed levels in the mouse ($\sigma_{\bar{z}} > 1 \times 10^{-2}$ [37]).

The model is naive in several other respects. First, gametes may often not be limiting and, as a result, the fitness of an individual may not be proportional to the probability of success of meiosis, such as assumed by the model thus far. A less-than-linear relation between fitness and success of meiosis can be modeled by increasing the number of meiosis that an individual can attempt before being declared sterile. Allowing for 5 attempts, thus mimicking a situation where an up to 80% reduction in the number of gametes would not substantially affect the reproductive success of individuals (sixth row of Table 2), we observe a higher functional diversity with a still reasonable level of erosion. The predicted reduction in meiosis success in

hemizygotes is also consistent with the ones observed empirically in mice ($\sim$ 2 to 5% [37]). However, the fitness now reacts more weakly to variation in the success of meiosis, and as a result, the model is running in an nearly-neutral regime, with a mean scaled selection coefficient acting on new alleles entering the population ($s_0$) smaller than $1/N_e = 10^{-5}$, such that the turnover of recombination landscapes is mostly driven by the neutral mutational turnover at the *PRDM9* locus. Although this could be seen as a possible working regime for the evolutionary dynamics of recombination given the high mutation rate at this locus, it is incompatible with the empirical support previously found for a positive selection acting on *PRDM9* [3, 24].

Alternatively, the version of the model considered thus far assumes that the total number of DSBs induced along the chromosome does not depend on the subsequent steps of the process. In reality, DSBs are tightly regulated, in a way that may entail a negative feedback inhibiting the creation of further DSBs once the chromosome has managed to synapse. For that reason, the mean number of DSBs per successful meiosis, which is what is empirically measured [13], may be substantially smaller than the maximum number of DSBs allowed before a meiocyte undergoes apoptosis. Yet, the success rate of meiosis depends on the maximum, not on the mean number. As a way to indirectly account for this, we performed a last simulation allowing for $d$ = 24 DSBs (instead of 8, last row of Table 2, Fig 9). Running the model under this configuration results in an evolutionary dynamics which is not in the eviction regime, with moderate levels of diversity ($D$ = 2.5), reasonable levels of erosion ($\bar{z} = 0.26$), and strong positive selection on new *PRDM9* alleles ($s_0 > 10^{-4}$). On the other hand, the haplo-insuffiency predicted for old alleles is now weaker than that observed empirically [37].

Altogether, these results show that the requirement of symmetric binding can result in a Red Queen maintaining erosion at moderate levels under empirically reasonable parameter values. On the other hand, there is only a narrow window for which eviction due to dosage is avoided and a sufficiently high *PRDM9* diversity is maintained, while still having haplo-insufficiency for old alleles and strong selection acting on *PRDM9*.

## Discussion

### Fundamental role of the symmetrical binding of PRDM9

The first studies on the function of *PRDM9* uncovered its role in targeting DSBs at specific sites, by its capacity to set histone marks to recruit the DSB machinery [11, 43–45]. More recently, it was discovered that this gene plays another important role during meiosis, namely, by facilitating chromosome pairing [1, 31]. In this process, the symmetrical binding of PRDM9 to its target sites plays a key role. By bringing to the chromosomal axis the sites on the homologue, PRDM9 presents them to the single-stranded DNA produced by the resection around the DSB, thereby facilitating the search for the complementary sequence (Fig 1). This second function of *PRDM9*, combined with the differential erosion of the target sites between sub-species (which is itself a direct consequence of the first function of *PRDM9*, the recruitment of the DSB machinery), was found to be sufficient for explaining most aspects of the hybrid sterility pattern [1, 31].

Here, we show that, similarly, these two aspects of the function of *PRDM9* during meiosis provide the necessary ingredients for explaining the selective force promoting the turnvover of *PRDM9* alleles in the context of a single population—thus giving a globally coherent picture bridging the gap between the molecular mechanisms and the evolutionary dynamics of recombination.

## Impact of gene dosage of *PRDM9* on the Red Queen dynamics

Gene dosage of *PRDM9* implies that a homozygote has a PRDM9 concentration for its allele which is twice that of each of the two alleles of a heterozygote. By the law of mass action, everything else being equal, increased dosage results in an increased occupancy of the targets and therefore an increased probability of symmetrical binding and a higher fertility. This clearly impacts the Red Queen dynamics for certain combinations of parameters, by acting against diversity and leading to a monomorphic regime characterized by the eviction of minor alleles (mostly found in a heterozygous state) by the currently dominant allele (mostly present in a homozygous state). The higher the selection coefficient associated to the homozygous advantage (captured here by $\sigma_0$), the stronger the effect against diversity. This impact of gene dosage on the evolutionary regime had been overseen thus far in the verbal and mathematical models of the Red Queen [18, 27], except in the more recent work of Baker *et al.* ([36], see below).

The impact of gene dosage is mitigated when PRDM9 is in high concentration in the cell. Indeed, a large concentration of PRDM9 makes the binding probability at a target of a given affinity less responsive to gene dosage (see S6 Fig), thus reducing the fertility gap between a homo- and a heterozygote (see S7 Fig). Based on these observations, we therefore hypothesize that the *PRDM9* expression level has been selected at a sufficiently high level so as to limit the impact of gene dosage. This is an interesting hypothesis that could be studied in the future.

## Comparison with Baker *et al.*'s model

In a recent article, Baker *et al.* [36] also propose a mechanistic model of the *PRDM9* Red Queen. Several of their conclusions agree with ours. Most notably, Baker *et al.* also recognize that the symmetrical binding of PRDM9 provides the key ingredient for mediating positive selection on PRDM9 in the context of the Red Queen, and they incorporate the consequences of gene dosage in their model. However, the two studies differ on some important points.

First, in our model, the amount of PRDM9 protein is not limiting (that is, most PRDM9 molecules are not bound to the target sites), whereas Baker *et al.* assume strong competition between targets for PRDM9 proteins (most molecules are bound). Of note, gene dosage and competition are two distinct aspects of the steady-state equilibrium of PRDM9 binding. Gene dosage is a direct consequence of the law of mass action (site occupancy is an increasing function of the free concentration of PRDM9, see S1 Fig), a phenomenon that can happen even when PRDM9 is not limiting, as shown here. Competition between targets, on the other hand, relates to the fact that the inactivation of some of the binding sites results in extra PRDM9 molecules available for other sites to bind, an effect that is negligible if PRDM9 is in excess (as in our model), but important otherwise (as in Baker *et al.*'s model [36]). Whether PRDM9 is limiting ultimately depends on its affinity for its targets, which is know to be high [46], but also on the total concentration of PRDM9 in the cell, which, to our knowledge, is still unknown, and finally, on the total number of targets in the genome. Altogether, the question thus still remains open. In any case, our model shows that competition between targets is not a necessary feature to drive the evolutionary dynamics of *PRDM9*.

The second difference concerns the affinity distribution. Baker *et al.* use a two-heat distribution. Here we use an exponential distribution (see S5 Fig), which is motivated by results from Chip-seq experiments [28], although with some uncertainty regarding the shape of the distribution in the low affinity range, which is not captured by Chip-seq. The use of these different affinity distributions has clear consequences on the behavior of models with respect to dosage. Indeed, increasing the gene dosage can have two opposing effects. On the one hand, a higher dosage increases the symmetrical binding at sites of intermediate affinity. On the other hand, it also results in more sites of low affinity being bound, and this, most often

asymmetrically. Depending on the balance between these two effects, different outcomes are obtained. In our case, where the affinity distribution has a moderate variance, the increased symmetrical binding of sites of intermediate affinity always wins. As a result, there is always an advantage to increasing dosage. That is, homozygotes always have an advantage over heterozygotes, which creates an eviction regime that tends to play against diversity. A contrario, in Baker's model, the two-heat affinity distribution results in a non-monotonous dependency, with a turning point, such that the dosage is in favor of homozygotes for young alleles, but in favor of heterozygotes for old alleles. This turning point is one potential solution to the problem of eviction. Here, we present an alternative solution, which is captured in our model by the statistics $\sigma_0\tau$. Intuitively, eviction does not take place if the homozygote advantage is sufficiently weak and erosion sufficiently rapid, such that ageing alleles are replaced before eviction has enough time to take place.

As it turns out, it is not so easy to find empirically reasonable parameter configurations such that the eviction regime is avoided, although this could be the consequence of other aspects of the biology of meiosis and reproduction being missed by the model. On the other hand, our model predicts that old alleles should be haplo-insufficient, thus in agreement with what is observed in the mouse (for allele B6 and C3H [37]). This haplo-insufficiency of old alleles is not observed systematically in Baker's *et al.* model, but only occasionally and in small populations.

Altogether it would be useful to empirically investigate the haplo-insufficiency for young and old alleles over a broader range of alleles and species, in order to determine whether a life-long homozygous advantage is systematic or occasional. More globally, it will be important to unravel the exact roles of affinity distribution, PRDM9 concentration and competition between targets in the evolutionary dynamic. Finally, all this raises the question of a possible evolution of the *PRDM9* expression level, the affinity (whether in its distribution or in its mean) and the number of targets recognized per allele.

## Current limitations and perspectives

In addition to those discussed above, the model introduced here has other limitations. First, in its current form, the model implements only limited variance among alleles in their strength at birth. Yet, in reality, some alleles are dominant over others [47], such that, in a heterozygote for a strong and a weak allele, DSBs are more often produced at the target sites of the stronger allele. Of note, although some dominance is expected to emerge purely as a consequence of erosion (with younger alleles being on average dominant over older ones), a substantial part of it appears to be instead related to intrinsic differences between alleles regardless of their age [15, 48]. If stronger alleles have an advantage over weaker alleles, either because they promote a higher binding symmetry or just because of their higher penetrance, then weaker alleles should be less likely to invade. As a result, the population would tend to be dominated by stronger alleles. In good approximation, this would amount to running the model under a lower effective mutation rate (now to be taken as the rate at which new functional and sufficiently strong *PRDM9* alleles are being produced by mutation) but otherwise would not fundamentally change the evolutionary dynamic.

Second, the model currently assumes that all DSBs are repaired with the homologue. In reality, some are repaired with the sister chromatid [15]. This simplifying assumption, however, should have a moderate impact on our conclusions, as it essentially amounts to a change in the rate of erosion, which can be accounted for by considering a lower mutation rate $v$ at the target sites. Our experiments suggest that small changes in $v$ do not fundamentally impact the dynamic.

Third, in addition to implementing the idea that symmetrical binding of PRDM9 is required for successful pairing of chromosomes, our model also more specifically assigns the (obligate and unique) CO to one of the DSBs occurring in symmetrically bound sites. In hindsight, this specific choice is not warranted by current empirical evidence, which only suggests that repair of DSBs with the homologue, and thus ultimately chromosome pairing, is facilitated by symmetrical binding of PRDM9, but this, independently of the choice of CO versus NCO among all DSBs. However, this specific feature of our model should have a minor impact on our conclusions. Choosing instead the CO uniformly at random among all DSBs (but still requiring at least one DSB in a symmetrically bound site) would merely change the patterns of linkage dissipation, and this, in a context where linkage is unlikely to be a major factor.

Although deserving more careful examination, the limitations just mentioned are therefore probably minor. Perhaps a more fundamental issue is to fully understand how the eviction regime induced by gene dosage effects can be avoided, so as to predict reasonable levels of *PRDM9* diversity, while having a Red Queen regime driven by strong positive selection on *PRDM9*. As it stands, the model appears to be stuck in a dilemma, such that either dosage has a substantial impact on fertility, but then eviction takes place and the model fails at explaining empirically observed levels of *PRDM9* diversity, or the effects of dosage are made weaker by assuming globally less stringent conditions for achieving a good fertility (such as allowing for an excess in PRDM9 protein, or in gametes or in DSBs), but then the fitness differences between old and new alleles also become small and the model approaches a nearly-neutral regime, in which the turnover of *PRDM9* alleles is primarily driven by the mutational turnover at the locus. Although the latter regime may not be so unreasonable from an evolutionary point of view, it fails at explaining the positive selection observed on *PRDM9*, or its haplo-insufficiency, depending on the detailed model configuration (last two rows of Table 2).

This dilemma is still in need of a robust and convincing explanation. In this respect, as mentioned above, the questions of the affinity distribution and the concentration of PRDM9 relative to the affinity of the target sites should be explored in depth. Ultimately, a more mechanistically motivated model of PRDM9 binding, accounting for the combinatorial sequence features susceptible to determine the affinity, could provide useful clues as to what would be a reasonable affinity distribution. It would also naturally lead to a more realistic model in which point mutations may only partially inactivate the targets [17], as opposed to completely inactivate them as in our current model.

Alternatively, the empirical regime of the Red Queen may possibly alternate between long nearly-neutral epochs, with occasional bouts of positive selection whenever the current alleles become too eroded. Such occasional epsiodes of positive selection could be sufficient to induce the empirically observed patterns of accelerated evolution of the zinc finger domain at the non-synonymous level. The impact of Hill-Robertson interference and its dissipation on the evolutionary dynamic of *PRDM9* could also contribute to maintaining a high diversity at the *PRDM9* locus, a point that may also need to be investigated. Finally, population structure could play a role, by maintaining different pools of alleles in different sub-populations connected by recurrent migration, thus resulting in a large diversity at the metapopulation level, and this, in spite of non-negligible gene dosage effects. Population structure is also pointing toward the other big question still in need of a model-based exploration: the potential role of *PRDM9* in hybrid sterility and speciation.

## Materials and methods

### The model

The simulation model is graphically summarized in Fig 2, and its key parameters are listed in Table 1. This model assumes of a population of *N* diploid individuals, whose genetic map is composed of a single chromosome. A *PRDM9* locus is located on this chromosome. Each *PRDM9* allele has a set of *h* binding sites, each of which has an intrinsic binding affinity for its cognate PRDM9 protein. The positions of binding sites along the chromosome are drawn uniformly at random and their affinities are drawn according to an exponential law of parameter $\bar{y}$ (see S5 Fig). Each target site is associated to a unique allele and different sites cannot overlap. In practice, this is implemented by encoding the chromosome as an array of *L* slots, such that, upon creation of a new *PRDM9* allele by mutation, each binding site of this allele chooses one of the available slots uniformly at random. Given the composition of the population of the current generation, the next generation, of the same population size, is generated as follows.

**Mutations.** The *PRDM9* locus mutates with a probability *u* per gene copy. Given a mutation, a new functional *PRDM9* allele is created to which are associated *h* new sites along the genome, according to the procedure just described. Next, each target site recognized by each allele currently present in the population mutates with a probability *v*. This type of mutation results in a complete inactivation of the target site. Of note, all target sites are assumed to be monomorphic for the active variant at the time of the birth of the corresponding *PRDM9* allele. As a result, the loss of target sites is entirely contributed by gene conversion acting on inactivating mutations that have occurred after the birth of the allele. Also, we neglect new target sites that might arise by mutation, which is reasonable since hotspot alleles newly arisen by mutation would be at a very high risk of being lost by conversion.

**Meiosis and reproduction.** A meiosis is attempted on a randomly selected individual, according to the following steps. Of note, the meiosis can fail (in which case we assume that the meiocyte undergoes apoptosis) for multiple reasons, all of which will be described below. In the simulations presented here, unless stated otherwise, whenever meiosis fails, then a new individual is chosen at random from the population and a new meiosis is attempted.

First, the two homologous chromosomes are replicated, thus creating a set of 4 chromatids. Then, PRDM9 binds to its target site according to an occupation probability determined by the chemical equilibrium. A binding site *i* has an affinity

$$K_i = \frac{[PS_i]}{[P]_{free}[S_i]} = \frac{x_i}{[P]_{free}(1 - x_i)}, \tag{7}$$

where $[P]_{free}$ is the free concentration of PRDM9 proteins in the cell and $[PS_i] = x_i$ and $[S_i] = 1 - x_i$ are the proportions of target site *i* (across meiocytes) which are occupied by PRDM9 or free, respectively. Thus:

$$x_i = \frac{[P]_{free}K_i}{1 + [P]_{free}K_i}. \tag{8}$$

We assume that PRDM9 is not limiting, meaning that most PRDM9 molecules are free $[P]_{free} \approx [P]_{tot}$ (where $[P]_{tot}$ is the total number of PRDM9 protein attributed to a given allele in the cell); there is therefore a total absence of competition between binding sites. Thus, if we define the rescaled affinity as $y_i = [P]_{tot}K_i$, then the occupancy probability can be re-written:

$$x_i = \frac{y_i}{1 + y_i}. \tag{9}$$

Based on this equation, for each target site, binding is randomly determined, by drawing a Bernoulli random variable of parameter $x_i$ for site $i$. Of note, at a given target locus, there are four instances of the binding site (one on each of the two sister chromatids for each of the two homologues), and binding is determined independently for each of those instances.

Once the occupation status of all binding sites has been determined, the total number $k$ of sites bound by PRDM9 over the four chromatids, is calculated. If $k = 0$, meiosis fails. Otherwise, each site bound by PRDM9 undergoes a DSB with a probability equal to $p = \min\left(1, \frac{d}{k}\right)$, with $d$ being a parameter of the model. In most experiments, we use $d = 6$. Thus, in the most frequent case where $k > d$, an average number of $d$ DSBs are being produced. This procedure aims to model the regulation of the total number of DSBs through the genome, which in mammals seems to be independent from PRDM9 binding [1, 13].

Next, the four chromatids are scanned for symmetrically bound sites, which the model assumes are essential for chromosome pairing (see Introduction). If no such site is detected, meiosis fails. Otherwise, one of the symmetrically bound sites is uniformly picked at random and becomes the initiation site for a CO. Thus, only one CO is performed per meiosis (and this, in order to model CO interference), and all other DSBs are repaired as NCO events. Note that in our model we do not allow for the possibility of DSB repair by the sister chromatid.

A successful meiosis therefore produces 4 chromatids, with exactly one CO between two of them, and some events of gene conversion at all sites that have undergone a DSB. Of note, in the presence of inactive versions of the binding sites (created by mutations, see above), these gene conversion events are effectively implementing the hotspot conversion paradox [16]. Finally one chromatid is randomly picked among the tetrad to become a gamete. The whole procedure is repeated, starting from the choice of a new individual until another gamete is obtained. Then these two gametes merge and create a new diploid individual of the next generation. All these steps, from the choice of an individual until the choice of a chromatid among the tetrad, are performed as many time as needed until the complete fill of the next generation.

**Gene dosage.** In the case where the gene dosage of PRDM9 is taken into account, it is assumed that, for a given *PRDM9* allele, a homozygote will produce twice the concentration of the corresponding protein compared to a heterozygote. By assuming that $[P]_{tot}^{homo} = 2[P]_{tot}^{het}$, the probability that a PRDM9 protein binds to a site $i$ of rescaled affinity $y_i$ is

$$x_i = \frac{cy_i}{1 + cy_i} \tag{10}$$

where $c = 1$ for a heterozygote and $c = 2$ for a homozygote.

## Summary statistics

Several summary statistics were monitored during the run of the simulation program. They were used, first, to evaluate the time until convergence to the equilibrium regime (burn-in). The burn-in was taken in excess, so as to be adapted to all simulation settings. In practice, the burn-in is set at 10000 generations over a total of 50, 000 generations. Second, the summary statistics were averaged over the entire run (burn-in excluded), thus giving a quantitative characterization of the equilibrium regime, as a function of the model parameters. The main statistics are the *PRDM9* diversity in the population ($D$), the mean proportion of binding sites that are still active per allele $\theta$, the mean symmetrical binding probability ($q$) and the mean fertility ($w$).

Diversity is defined as the effective number of *PRDM9* alleles, or in other words as the inverse of the homozygosity [27]. At time *t*:

$$D_t = \frac{1}{\sum_i f_{i,t}^2} \tag{11}$$

where $f_{i,t}$ is the frequency of allele *i* at time *t*. With this definition, when K alleles segregate each at frequency $\frac{1}{K}$ in the population, the diversity *D* is equal to *K*. Averaging Eq 11 over the simulation run gives the average diversity *D*.

The mean proportion of binding sites that are still active per allele is defined as:

$$\theta_t = \sum_i f_{i,t}\theta_{i,t} \tag{12}$$

where $\theta_{i,t}$ is the proportion of sites that are still active for allele *i* at time *t*. Averaging $\theta_t$ over the simulation trajectory gives $\bar{\theta}$.

The probability of symmetrical binding *q* corresponds to the mean probability of having a PRDM9 protein bound on at least one of the two chromatids of the homologous chromosome at a certain position, given that a DSB has occurred at this very position. In the case of a homozygous individual for allele *i*, this quantity can be obtained analytically for a given complete diploid genotype and is given by:

$$q_{i,t}^{hom} = \frac{2 < x^2 > - < x^3 >}{< x >}, \tag{13}$$

where $x = \frac{cy}{1+cy}$ is the occupancy of a binding site of affinity *y* and, for any *k*, $< x^k >$ is the mean of $x^k$ over all sites, taking into account their affinity distribution. In the case of a heterozygous individual, with two *PRDM9* alleles *i* and *j*, the symmetrical binding rate is:

$$q_{i,j,t}^{het} = \frac{2 < x^2 >_i - < x^3 >_i + 2 < x^2 >_j - < x^3 >_j}{< x >_i + < x >_j}, \tag{14}$$

where the subscripts *i* and *j* correspond to averages over sites of allele *i* and *j*, respectively. For allele *i*, these statistics were then averaged over all diploid backgrounds for this allele (either homo- or heterozygotes), thus against each allele *j* present in the population at a given generation, yielding the mean symmetrical binding rate for allele *i* at time *t*, $q_{i,t}$. Finally, this was averaged over all alleles (weighted by their frequency), and over the simulation run, yielding the overall population-level equilibrium mean value of *q*.

For a given genotype, the fertility can be computed analytically. Here we assumed that fertility in proportional to the rate of success of meiosis. Thus, it is equivalent to 1-(the mean probability of failure of meiosis) characterized by the absence of a DSB in a symmetric site. In turn, the number of DSBs in a symmetric site follows a Poisson law with parameter *dq* where *d* is the average number of DSBs per meiocyte and *q* is the mean probability of symmetrical binding for this allele. In the case of a homozygous individual for allele *i*, this quantity can be obtained analytically for a given complete diploid genotype and is given by:

$$w_{i,t}^{hom} = 1 - e^{-dq_{i,t}^{hom}} \tag{15}$$

In the case of a heterozygous individual, with two *PRDM9* alleles *i* and *j*, the fertility rate is:

$$w_{i,j,t}^{het} = 1 - e^{-dq_{i,j,t}^{het}} \tag{16}$$

where the subscripts $i$ and $j$ correspond to averages over sites of allele $i$ and $j$, respectively. As for the symmetrical binding rate, for each allele $i$, these statistics were averaged over all diploid backgrounds for this allele (either homo- or heterozygotes), against all other alleles present in the population at a given generation, yielding the fertility rate for allele $i$ at time $t$, $w_{i,t}$. Finally, this was averaged over all alleles (weighted by their frequency), and over the simulation run, yielding the overall population-level equilibrium mean value of $w$.

The haplo-insufficiency coefficient for an allele $i$ at time $t$ $\sigma_{i,t}$ is defined as the relative difference in fertility between homozygotes ($w^{hom}$) and hemizygotes ($w^{hemi}$) for this specific allele:

$$\sigma_{i,t} = \frac{w_{i,t}^{hom} - w_{i,t}^{hemi}}{w_{i,t}^{hemi}} . \tag{17}$$

Here, the hemizygote fertility is given by:

$$w_{i,t}^{hemi} = 1 - e^{-dd_{i,t}^{hemi}} \tag{18}$$

and

$$q_{i,t}^{hemi} = \frac{2 <x^2> - <x^3>}{<x>}, \tag{19}$$

Of note, although Eqs 19 and 13 look identical, the site occupancies $x$ are different, owing to the differing concentrations of the protein corresponding to allele $i$ when in one or two copies. This can again be averaged over all alleles in the population and over the run, yielding $\bar{\sigma}$.

The haplo-insufficiency coefficient of a new allele, $\sigma_0$, is defined similarly as for segregating alleles (Eq 17), although its actual computation proceeds differently. Taking advantage of the fact that this coefficient does not entail any average over the genetic backgrounds segregating in the population, it is obtained independently of the simulation run, by randomly creating new alleles (and their corresponding arrays of binding sites over the genome) and then computing the symmetrical binding rate, and then the fertility, assuming one or two copies for this allele. This is repeated 100 times, and the average over these 100 alleles yields an estimate of $\sigma_0$.

Similarly, the selection coefficient for new alleles arising in the population, $s_0$, is defined, and computed, as the difference between the log fitness of a new allele (created on the fly and combined with all genetic backgrounds segregating in the population) and the log fitness averaged over the whole population. The resulting time-dependent estimate is then averaged over the simulation run. The complete derivation is available in Supplementary materials S1 Appendix.

## Scaling experiments

In order to visualize how the equilibrium regime varies according to the model parameters, first, a central parameter configuration was chosen, which will represent a fixed reference across all scaling experiments. Then only one parameter at a time (or two for bi-dimensional scaling) is systematically varied over a range of regularly spaced values on both sides of the reference configuration. This parameter is fixed along the entire simulation and is variable only between simulations. For each parameter value over this range, a complete simulation is run. Once the equilibrium is reached for a given simulation setting, the summary statistics described above are averaged over the entire trajectory, excluding the burn-in. These mean equilibrium values, which characterize and quantify the stationary regime of the model, were finally plotted as a function of the varying parameter(s).

The central parameters were chosen as follows. First, the parameters of population size $N$ was set to 5, 000, corresponding to the maximum population size that can be afforded computationally. Then the number of sites recognized as target sites by each PDRM9 allele's protein was set to $h = 400$ in order to get closer to the average number of target sites found on the smallest chromosome of the mouse (around 800 sites) while limiting the memory requirements. The mean for the affinity distribution of the target was set to $\bar{y} = 6$. The parameter $d$ representing the number average of DSBs in a meiocyte is set at 6 which corresponds to the approximate number of DSBs found on the smallest chromosome of the mouse. For unidimensional scaling, the parameter $v$, the mutation rate at target sites, was set to $2 \times 10^{-6}$ and the parameter $u$ was set to $5 \times 10^{-5}$. For the bi-dimensional scaling of $d$ and $\bar{y}$, $u = 2 \times 10^{-4}$ and $v = 5 \times 10^{-5}$.

## Supporting information

**S1 Fig. A simulation trajectory under a polymorphic regime (u = 5 × 10⁻⁴, N = 5 × 10³ and v = 5 × 10⁻⁵) with same scale as 3.** In all panels, each color corresponds to a different allele. Note that a given color can be reassigned to a new allele later in the simulation. Successive panels represent the variation through time of (A) the frequency of each *PRDM9* allele and its corresponding (B) the proportion of active sites, (C) the mean affinity of active sites, (D) the probability of symmetrical binding and (E) the fertility. The thick line singles out the trajectory of a typical allele.
(TIF)

**S2 Fig. Two simulation trajectories under the control model allowing for chromosome pairing and success of meiosis without requiring symmetrical binding of PRDM9.** (A) and (B) correspond to the monomorphic regime (as in 3, **u = 5 × 10⁻⁶, N = 5 × 10³** and **v = 5 × 10⁻⁵**), while (C) and (D) correspond to the polymorphic regime (as in 4, **u = 5 × 10⁻⁴, N = 5 × 10³** and **v = 5 × 10⁻⁵**). In all panels, each color corresponds to a different allele. Note that a given color can be reassigned to a new allele later in the simulation. Successive panels represent the variation through time of (A) and (C) the frequency of each *PRDM9* allele and (B) and (D) its corresponding proportion of active sites.
(TIF)

**S3 Fig. A simulation trajectory with genetic dosage and under the control model allowing for chromosome pairing and success of meiosis without requiring symmetrical binding of PRDM9.** The simulation was run with **u = 5 × 10⁻⁴, N = 5 × 10³** and **v = 5 × 10⁻⁵**. In all panels, each color corresponds to a different allele. Note that a given color can be reassigned to a new allele later in the simulation. Successive panels represent the variation through time of (A) the frequency of each *PRDM9* allele and its corresponding (B) the proportion of active sites.
(TIF)

**S4 Fig. Scaling of key summary statistics at equilibrium, as a function of the mutation rate u at the *PRDM9* locus and the mutation rate v at the target sites (N = 5 × 10³).** The statistics are: the *PRDM9* diversity $D$ as a function of $u$ (A) and $v$ (D); the mean erosion $\bar{z}$ (i.e. the mean fraction of target sites that have been inactivated) at equilibrium as a function of $u$ (B) and $v$ (E); the mean fertility of the population $\bar{w}$ as a function of $u$ (C) and $v$ (F). On each graph, the mean (blue line) and standard variation (blue vertical bars) over a simulation are displayed against the prediction of the analytical approximation (orange line). The area colored in green corresponds to the range of parameters for which the analytical model verifies the assumptions of a high diversity ($1 < 4Nu < 100$), a low erosion ($\bar{z} < 0.5$) and strong selection on new *PRDM9* alleles ($4Ns_0 > 3$). The analytical approximations presented here are plotted on panels

A to F (orange curves), against the results obtained directly using the simulation program (blue curves). The model and the analytical approximation give qualitatively similar results in the range of parameters validating all the conditions (in practice, we consider that the analytical results should be valid in the following intervals for the model parameters: $1 < 4Nu < 100$, $\bar{z} < 0.5$ and $4Ns_0 > 3$). Concerning *PRDM9* diversity, substantial differences are observed between the simulation results and the analytical approximations, up to a factor of 10, in the scaling of $u$ (panel A). However, the nature of the regime, polymorphic (many alleles segregating at the same time in the population) or monomorphic (only one allele present in the population at a time), is correctly predicted. In particular, we can say that the nature of the regime is directly and mostly determined by $Nu$ and the level of erosion has almost no influence (panel D). Finally, the analytical approximations are less accurate for low and high $u$ or high $v$. These correspond to strong erosion regimes (low $u$ and high $v$) or to regimes with weak selection (high $u$), for which the assumption of the analytical developments are not met.
(TIF)

**S5 Fig. Exponential law for affinity distribution (with mean *y* = 0.2).** The continuous blue line corresponds to the affinity distribution for young alleles and the dotted red line corresponds to the affinity distribution for old alleles.
(TIF)

**S6 Fig. Binding probability as a function of the concentration of free PRDM9 protein molecules for homozygotes and heterozygotes (mean affinity y = 0.6).** The binding probability for homozygotes is always higher than that for heterozygotes, but the difference between them decreases when the free PRDM9 concentration increases.
(TIF)

**S7 Fig. Selection coefficient associated to gene dosage ($\sigma_0$) as a function of the concentration of PRDM9 in the cell and the mean affinity of the target sites ($\bar{y}$).**
(TIF)

**S1 Appendix. Derivation of the analytical approximation to the equilibrium regime of the Red Queen.**
(PDF)

**S1 Text. Empirical *PRDM9* diversity in mouse subspecies.**
(PDF)

## Acknowledgments

We would like to thank Corinne Grey and Frédéric Baudat for their comments on the manuscript, as well as Zachary Baker for helpful discussions. All simulations of this work were performed using the computing facilities of the CC LBBE/PRABI.

## Author Contributions

**Conceptualization:** Alice Genestier, Nicolas Lartillot.

**Data curation:** Alice Genestier.

**Formal analysis:** Alice Genestier.

**Funding acquisition:** Laurent Duret, Nicolas Lartillot.

**Investigation:** Alice Genestier.

**Methodology:** Alice Genestier.

**Project administration:** Laurent Duret, Nicolas Lartillot.

**Software:** Alice Genestier.

**Supervision:** Laurent Duret, Nicolas Lartillot.

**Visualization:** Alice Genestier.

**Writing – original draft:** Alice Genestier.

**Writing – review & editing:** Alice Genestier, Laurent Duret, Nicolas Lartillot.

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
