## [Decision Letter · Decision Letter 0]

5 Jul 2023

Dear Dr Genestier,

Thank you very much for submitting your Research Article entitled 'Bridging the gap between the evolutionary dynamics and the molecular mechanisms of meiosis: a model based exploration of the PRDM9 intra-genomic Red Queen' to PLOS Genetics.

The manuscript was fully evaluated at the editorial level and by independent peer reviewers. All the reviewers appreciate the modelling approach to understand the behaviour and evolution of PRDM9, and find the new insights this results in interesting - particularly the impact of dosage effects, interacting with fitness, on the evolutionary dynamics of the protein and hence the recombination landscape. 

However, the reviewers also raised some substantial concerns about the current manuscript. Based on the reviews, we will not be able to accept this version of the manuscript, but we would be willing to review a much-revised version. We cannot, of course, promise publication at that time.

Should you decide to revise the manuscript for further consideration here, your revisions should address the specific points made by each reviewer. We will also require a detailed list of your responses to the review comments and a description of the changes you have made in the manuscript. In particular, please carefully address the following points raised in the review process:

1)    Concepts. All of the reviewers noted that several concepts should be defined/described to enable them to be fully accessible for non-specialist readers. In particular, please ensure the manuscript includes clear definitions of (i) the “erosion” of hotspots (which I think here means their loss via the action of the hotspot paradox), and (ii) the Red Queen hypothesis (in the general sense), used throughout this work. It would also be helpful to (iii) explain that in the case of interest, hotspots are positioned due to their being strong binding sites of PRDM9, (iv) explain the significance of the minisatellite nature of the PRDM9 ZF array, in allowing the potential for the rapid accumulation of new, functional alleles with completely different hotspots, and for a high mutation rate. (v) Explain the “Wright-Fisher” model (briefly). (vi) make sure “symmetrical binding” is well defined.

2)    Dominance of PRDM9 alleles (reviewer 1). Please explain if this is accounted for in the model and/or the implications of dominance of some PRDM9 alleles which can be strong in nature, and might of course evolve for model-based predictions of PRDM9 evolution. [NB dominance does not appear to be a purely emergent phenomenon from erosion, because in several studies it is observed even for pairs of alleles whose binding sites have not been eroded in the genome of interest. For example, does the existence of dominance provide a selective advantage to more dominant newly-arising alleles.]

3)    Resolution of double strand breaks (DSBs) – please explain the implications of repair via crossovers, non-crossovers (which contribute to motif erosion) or sister chromatid repair (which likely doesn’t) to the model, and add to the discussion the possibility that at heterozygous hotspot loci the proportion of DSBs repairing using the sister chromatid might be higher.

4)    Nomenclature, which the reviewers note is not consistent/correct in various places: several examples are in the abstract. In particular, please check/use standard nomenclature for e.g. gene names: use upper case non-italicised letters to refer to a protein, and italicised letters to refer to the gene/DNA sequence, capitalising the whole name for human gene names or only the first letter for the mouse gene, and make it clear to the reader whether the gene (natural e.g. if considering alleles or mutations to alleles)/protein (natural e.g. if considering binding properties, etc etc.) is being referred to.

5)    All the reviewers highlighted typos that should be fixed, suggest removing unnecessary abbreviations (e.g. “dBGC”), and also suggest that paragraphs might be shortened in a number of places.

6)    Please address the other points made by individual reviewers in full, including in particular suggestions for figure modifications by reviewers 2 and 3, and the suggestion of reviewer 1 to discuss implications of potential migration/population structure for the models.

7)   Reviewer 2 describes a relaxation of the assumption that PRDM9 dosage is not limiting as being a component of the model incorporating dosage, and this seems a logical description of the later model. In the discussion, though, this is highlighted as a difference with the Baker et al. approach: “in our model, the amount of PRDM9 protein is not limiting”, going on to say that the model of this manuscript does not include competition between PRDM9 targets. Can you either reword or add an explanation of this fairly subtle difference, whereby gene dosage can have an impact (suggesting more PRDM9 corresponds to more binding) but without there being actual competition between binding sites? Further, please specify in what sense can this correspond to gene dosage being “non-limiting”, if higher dosage of an allele means higher binding to sites and higher fitness? 

 8)  The manuscript contains the line “this step of meiosis is critical, since it is what will allow the two homologues to associate, realize a CO and form their synapsis”. Perhaps reorder to place synapsis in front of CO, because at present this reads as though CO is required for or precedes synapsis, but this is not the case - rather COs are linked most closely to (avoiding) aneuploidy. DSBs are likely required for synapsis in the species of interest here; it is further very likely that homologue engagement (which may involve either CO or NCO events) is also essential. There are a few other places where this potential misunderstanding also occurs, e.g. line 174. (Available evidence from the Forejt lab’s papers also suggest ~20Mb of homozygosity, enough for ~1 DSB to reliably occur, but not for a CO, is enough to restore synapsis of a chromosome.)

If you decide to revise the manuscript for further consideration at PLOS Genetics, please aim to resubmit within the next 60 days, unless it will take extra time to address the concerns of the reviewers, in which case we would appreciate an expected resubmission date by email to plosgenetics@plos.org.

To enhance the reproducibility of your resu

---

## [Decision Letter · Decision Letter 1]

5 Dec 2023

Dear Dr Genestier,

Thank you very much for submitting your Research Article entitled 'Bridging the gap between the evolutionary dynamics and the molecular mechanisms of meiosis: a model based exploration of the PRDM9 intra-genomic Red Queen' to PLOS Genetics.

The manuscript was fully evaluated at the editorial level and by an independent peer reviewer. The reviewer appreciated the attention to an important problem, but raised some substantial concerns about the current manuscript. Based on the reviews, we will not be able to accept this version of the manuscript, but we would be willing to consider a much-revised version. We cannot, of course, promise publication at that time.

While the revised manuscript thoroughly addresses many of the specific points made by the original reviewers, there remain serious reviewer concerns that a particular, and quite central, aspect of the modelling reduces the impact and usefulness of the results at present: the use and weighting attached to the “no-dosage” model whereby PRDM9 dosage does not impact fertility. Balancing this against the other two reviewer’s positive opinions - that the comparison of models brings insights - and even though no-dosage is not used for the whole paper, it still does at present cloud the major insights all reviewers agree this work would otherwise bring. First, there is an issue of how – or if – the model is justified biologically. Second, the reviewer feels much too much weight is given to this model in the manuscript main text and figures, especially if it is seen as a “straw man” hypothesis. It is essential that these points are addressed. There are of course, out of necessity many factors that cannot be taken into account in a model-based approach. None of the reviewers believe that, for example, it is essential to model the impact of numbers/sizes of chromosomes, or multiplicative effects of asynapsis on infertility due to e.g. cytoplasmic bridges between cells, to gain useful insights on evolutionary dynamics from “red queen” like dynamics. However, dosage is more basic than these and has more potential for useful/reasonable modelling.

In justifying the “no-dosage” model, the existing approach to say “Although empirically questionable, this assumption offers a simpler basis for understanding key features of the model and of the resulting evolutionary dynamics” was highly ambiguous to the reviewer. Similar is “Note that the parameter values used here are not meant to be empirically relevant”. These are insufficiently strong and indeed suggest this model *might* be right or would be correct with appropriate parameter choices. Then, on page 12 the statement “the previous results were obtained with a model assuming the same concentration of the PRDM9 protein product of a given allele in individuals that are either homozygous or heterozygous for this allele” is worrying to the reader, because putting it this way implies the prior work – already read at this point - does not in fact reflect any reasonable biological reality. To be clearer to readers, improved justification of the use of this model needs to be added. First, is there any binding or – perhaps more likely – fertility model in which the “no-dosage” approach would be correct (see point 2), as suggested by “questionable”? OR is it a key boundary case, and e.g. we expect the real world to lie between the no-dosage and completely dosage dependent extremes. As an example suggestion, perhaps this might happen if there are timing issues in meiosis – which is almost certain in reality. A time window for synapsis might allow partial recovery of fertility from a lower level of instantaneous binding in a heterozygote by allowing for synapsis if binding occurs at any point within the time range. This makes synapsis less sensitive to hotspot "heat", and so dosage, but still sensitive to complete hotspot loss - might this allow for a spectrum of heterozygote disadvantage.  It might then be appropriate to simply view dosage impact as a parameter, and evaluate the edge cases. Secondly, if not realistic, then given that later comparisons with an obviously unrealistic model are not of immediately clear value, then explain more precisely why are we looking at this model. If it is essential to understand this model, in order to build or understand more realistic models, this should be stated. If this model is more amenable to the theoretical calculations this should similarly be stated – but ultimately, biological relevance of the theoretical calculations should still be justified.In terms of weighting, I concur with the reviewer that the main text figures and descriptions currently focus too heavily on the apparently more unrealistic model at the expense of the more realistic models. Currently I believe Figure 3, Figures 4 and 5., and also Figure 6A-C relate only to the no-dosage model. Only in Figure 6D-F is the more realistic model considered. Then Figures 7 and 8 use this model, attempting to fit real-world parameters more precisely. To address the reviewer's main remaining concern, consider reversing the order of discussion to only consider the no-dosage model **after** the more realistic model. This would allow you to motivate the no-dosage model as a way of allowing the evolution of higher diversity levels, that are often seen in nature - and then allowing discussion of how this feature might occur biologically, at least via reduced dosage. In any case, please ensure that the more realistic model gains at least as much representation in the Figures as the no-dosage model, e.g. by moving most or all of these earlier figures (3-5) to the Supplementary material, or deleting less important ones entirely. Existing literature relevant to dosage models. In relevant meiotic cells, the “heat” parameters of your models may in the simplest models be thought of as driven by two other parameters: the “kon” rate at which an individual PRDM9 molecule binds to a single DNA site, and the “koff” rate at which they dissociate from that DNA site. Does the “non-limiting” model of your work correspond to a high value of koff for example, relative to kon? Attempts have been made to estimate PRDM9 behaviour in terms of e.g. koff; see Striedner et al. 2016 for example. Alongside the Paigen lab estimated values for PRDM9 copies per cell, other work from the Forejt lab has suggested that a single DSB at a symmetrically bound site might be enough for a chromosome to synapse (~20Mb of homozygosity being enough for asynapsis to be largely relieved in practice). Please consider citing these prior studies or other relevant ones – and definitely explain how your models fit in with this literature, or how the literature suggests future amendments.Please read the review of the revised manuscript carefully and address all the other specific points raised. In particular, please carefully consider the points made under “Inadequate testing of the hypothesis that symmetrical pairing promotes Red Queen dynamics”. I agree with the reviewer that use of the dosage model is important here. I would also highlight that existing literature suggests higher rates of crossover AND noncrossover events at symmetric hotspots. Do you really need to focus on CO events? There are reasons to believe synapsis might not be closely linked to CO (vs NCO) events, but instead more directly to symmetric binding of any DSB.Please correct the following minor issues with the revision not highlighted by the reviewer: please define what w is a function of (i.e. two parameters) in Table 1. Also explain w_hom and w_het and their relationship with w, w*, w bar and the various parameters – at the moment, differing versions of these functions appear across the tables, methods and supplement and it is not always immediately clear how they relate to one another and the other parameters. Also, in Table 1 title, fix the typo “Descritption”.

If you decide to revise the manuscript for further consideration at PLOS Genetics, please aim to resubmit within the next 60 days, unless it will take extra time to address the concerns of the reviewers, in which case we would appreciate an expected resubmission date by email to plosgenetics@plos.org.

We are sorry that we cannot be more positive about your manuscript at this stage. Please do not hesitate to contact us if you have any concerns or questions.

Yours sincerely,

Simon Myers

Guest Editor

PLOS Genetics

Gregory P. Copenhaver

Editor-in-Chief

PLOS Genetics

Reviewer's Responses to Questions

**Comments to the Authors:**

Reviewer #2: Uploaded as an attachment

**Have all data underlying the figures and results presented in the manuscript been provided?**

Reviewer #2: Yes

PLOS authors have the option to publish the peer review history of their article (what does this mean?). If published, this will include your full peer review and any attached files.

Reviewer #2: **Yes: **Rosemary Redfield

---

## [Editor Report · Decision Letter 2]

26 Apr 2024

Dear Dr Genestier,

We are pleased to inform you that your manuscript entitled "Bridging the gap between the evolutionary dynamics and the molecular mechanisms of meiosis: a model based exploration of the PRDM9 intra-genomic Red Queen" has been editorially accepted for publication in PLOS Genetics. Congratulations!

Before your submission can be formally accepted and sent to production you will need to address the final minor comments from the Editor (see below), and to complete our formatting changes, which you will receive in a follow up email. Please be aware that it may take several days for you to receive this email; during this time no action is required by you. Please note: the accept date on your published article will reflect the date of this provisional acceptance, but your manuscript will not be scheduled for publication until the required changes have been made.

Yours sincerely,

Simon Myers

Guest Editor

PLOS Genetics

Gregory Copenhaver

Section Editor

PLOS Genetics

Editor comments:

We thank the authors for their careful revision of this extremely interesting manuscript, which means it is now suitable for acceptance in PLoS Genetics.

The manuscript is improved in key ways, addressing the key concerns of the associate editor and reviewer. In particular:

1) The description of the no-dosage model as a pathway to understanding both the simulation and theoretical results is now much more clearly motivated, and the revised text, figures and tables suitably remove the relative dominance of this model in earlier versions of the manuscript. I do believe that this model is important to consider, because through this it is (now) clear that dosage is likely essential for the observed real-world behaviour of hotspot evolution - in particular in providing a critical barrier for the emergence of new alleles that can result in greater hotspot decay and reduced diversity (both likely critical in e.g. hybrid sterility), than if it were not present.

2) There are new simulations presented of a control model without a symmetry requirement, as suggested by the reviewer, showing that hotspots do still evolve in this scenario, but much less rapidly.

3) There are many places where the text has been clarified, made more concise and restructured, and the figures and tables have been thoroughly revised.

I noticed several remaining minor issues and typos that should be addressed prior to publication, with the clarification of lines 440-467 perhaps being the most substantive issue because it has implications for understanding of some of the main results of the paper.

Figure S3 and S4 do not seem to cited anywhere (but might overlap with the missing citations below).

The format of referring to supplementary figures is inconsistent throughout the text, and should be made consistent.

Line 272 – does this in fact refer to/as well as Fig S3?

Line 323 – no Figure named

Line 341 – no Figure named

Line 383 should be “..relies on several…”

Line 399 should refer to the appropriate derivation, perhaps in appendix S1

Line 415 “hemizgote” is a typo

Lines 440 to 467 are confusing to read, but this can be simply addressed I think:

(i) add references to specific results in the Appendix or other parts of derivation justifying the points. The links are non-obvious in some cases, because the derivation in Appendix S1 does not, for example, dwell on sigma_0 \\times \\tau.

(ii) add a one-line interpretation of sigma_0 \\times \\tau – e.g it might relate to the disadvantage faced by newly arising alleles, relative to the amount of erosion expected in the population to overcome this, so if large it intuitively points towards a strong impact of dosage?

(iii) there are apparent (not necessarily actual) contradictions in this section. Please fix these or if not incorrect, avoid the impression of contradictions. It is stated on line 441 “there should be enough time between successive invasions for eviction to take place. Thus, tau should be small”. The first line implies time tau being large – the second sentence then says this same time should be small. On line 448, it is stated that sigma_0 \\tau should be large for a “qualitative change induced by gene dosage” and it should be explained what this means if \\tau should simultaneously be small – so sigma_0 is very large? On line 464, sigma_0 \\tau must be small for “a change of regime” – apparently now referring to this quantity being small….. This might make sense somehow, or there might be typos, but it certainly seems quite confounding to the reader at present,

I struggled to reconcile my interpretation of your results with all these statements. My intuition is that dosage models alter the dynamics by preventing invasion, via erecting a barrier to newly arising (so heterozygous) alleles, provided that existing alleles are sufficiently monomorphic to commonly exist in the homozygous state. To change behaviour then, introducing dosage should alter things in a setting where – absent dosage – there is some erosion but not so much as to overcome this barrier, and where the mutation rate is high enough to not be monomorphic, so dosage has a behaviour to change. However if this understanding is correct, it would seem to contradict some lines of the section, e.g. it would suggest higher het. advantage would increase impact of dosage, so would higher tau, so sigma_0 tau should be large, while tau should perhaps be small for polymorphism to occur absent dosage.

Line 469 “substantially” typo

Line 471 talks about a “sufficiently short time between invasions” – see above comments on lines 440 to 467.

Line 804 x_1 should be x_i

Line 807 is [P]_free defined already? similarly [P]_tot

Line 866 typo “occured”

Table 2 legend refers to \\bar{z} but column header is 1-\\bar{theta}, and refers to \\sigma_{\\bar{z}} but column header is just \\bar{\\sigma}. The legend also says “fixed values….(d=8,” but this is not in fact a fixed value throughout the table.

Appendix S1: PRDM9/Prdm9 are used interchangeably and italicisation might also be incorrect in places – please make consistent.

In appendix S1, is it correct/more efficient to note that equation (12) implies z=2 \\bar{z} and so with (16) we obtain equation (19) immediately (certainly this results in the same endpoint)?

**Data Deposition**

http://datadryad.org/submit?journalID=pgenetics&manu=PGENETICS-D-23-00420R2

**Press Queries**

---

## [Editor Report · Acceptance letter]

16 May 2024

PGENETICS-D-23-00420R2 

Bridging the gap between the evolutionary dynamics and the molecular mechanisms of meiosis: a model based exploration of the *PRDM9* intra-genomic Red Queen 

Dear Dr Genestier, 

We are pleased to inform you that your manuscript entitled "Bridging the gap between the evolutionary dynamics and the molecular mechanisms of meiosis: a model based exploration of the *PRDM9* intra-genomic Red Queen" has been formally accepted for publication in PLOS Genetics! Your manuscript is now with our production department and you will be notified of the publication date in due course.

With kind regards,

Anita Estes

PLOS Genetics

On behalf of:
